# Quantifying Learning Guarantees for Convex but Inconsistent Surrogates

**Kirill Struminsky**
NRU HSE,[*] Moscow, Russia

**Simon Lacoste-Julien**[†]
MILA and DIRO
Université de Montréal, Canada

**Anton Osokin**
NRU HSE,[*‡] Moscow, Russia
Skoltech,[§] Moscow, Russia

## Abstract

We study consistency properties of machine learning methods based on minimizing convex surrogates. We extend the recent framework of Osokin et al. [14] for the quantitative analysis of consistency properties to the case of inconsistent surrogates. Our key technical contribution consists in a new lower bound on the calibration function for the quadratic surrogate, which is non-trivial (not always zero) for inconsistent cases. The new bound allows to quantify the level of inconsistency of the setting and shows how learning with inconsistent surrogates can have guarantees on sample complexity and optimization difficulty. We apply our theory to two concrete cases: multi-class classification with the tree-structured loss and ranking with the mean average precision loss. The results show the approximation-computation trade-offs caused by inconsistent surrogates and their potential benefits.

## 1 Introduction

Consistency is a desirable property of any statistical estimator, which informally means that in the limit of infinite data, the estimator converges to the correct quantity. In the context of machine learning algorithms based on surrogate loss minimization, we usually use the notion of Fisher consistency, which means that the exact minimization of the expected surrogate loss leads to the exact minimization of the actual task loss. It can be shown that Fisher consistency is closely related to the question of infinite-sample consistency (a.k.a. classification calibration) of the surrogate loss with respect to the task loss (see [2, 17] for a detailed review).

The property of infinite-sample consistency (which we will refer to as simply consistency) shows that the minimization of a particular surrogate is the right problem to solve, but it becomes especially attractive when one can actually minimize the surrogate, which is the case, e.g, when the surrogate is convex. Consistency of convex surrogates has been the central question of many studies for such problems as binary classification [2, 24, 19], multi-class classification [23, 21, 1, 17], ranking [11, 4, 5, 18, 15] and, more recently, structured prediction [7, 14].

Recently, Osokin et al. [14] have pinpointed that in some cases minimizing a consistent convex surrogate might be not sufficient for efficient learning. In particular, when the number of possible predictions is large (which is typically the case in the settings of structured prediction and ranking) reaching adequately small value of the expected task loss can be practically impossible, because one would need to optimize the surrogate to high accuracy, which requires an intractable number of iterations of the optimization algorithm.

---

[*]National Research University Higher School of Economics

[†]CIFAR Fellow

[‡]Samsung-HSE Joint Lab

[§]Skolkovo Institute of Science and Technology

It also turns out [14] that the possibility of efficient learning is related to the structure of the task loss. The 0-1 loss, which does not make distinction between different kinds of errors, shows the worst case behavior. However, more structured losses, e.g., the Hamming distance between sequence labelings, allow efficient learning if the score vector is designed appropriately (for the Hamming distance, the score for a complete configuration should be decomposable into the sum of scores for individual elements).

However, the analysis of Osokin et al. [14] gives non-trivial conclusions only for consistent surrogates. At the same time it is known that inconsistent surrogates often work well in practice (for example, the Crammer-Singer formulation of multi-class SVM [8], or its generalization structured SVM [20, 22]). There have indeed been several works to analyze inconsistent surrogates [12, 18, 5, 14], but they usually end the story with proving that some surrogate (or a family or surrogates) is not consistent.

**Contributions.** In this work, we look at the problem from a more quantitative angle and analyze to which extent inconsistent surrogates can be useful for learning. We focus on the same setting as [14] and generalize their results to the case of inconsistent surrogates (their bounds are trivial for these cases) to be able to draw non-trivial conclusions. The main technical contribution consists in a tighter lower bound on the calibration function (Theorem 3), which is strictly more general than the bound of [14]. Notably, our bound is non-trivial in the case when the surrogate is not consistent and quantifies to which degree learning with inconsistent surrogates is possible. We further study the behavior of our bound in two practical scenarios: multi-class classification with a tree-structured loss and ranking with the mean average precision (mAP) loss. For the tree-structured loss, our bound shows that there can be a trade-off between the best achievable accuracy and the speed of convergence. For the mAP loss, we use our tools to study the (non-)existence of consistent convex surrogates of a particular dimension (an important issue for the task of ranking [11, 4, 5, 18, 17]) and quantify to which extent our quadratic surrogate with the score vector of insufficient dimension is consistent.

This paper is organized as follows. First, we introduce the setting we work with in Section 2 and review the key results of [14] in Section 3. In Section 4, we prove our main theoretical result, which is a new lower bound on the calibration function. In Section 5, we analyze the behavior of our bound for the two different settings: multi-class classification and ranking (the mean average precision loss). Finally, we review the related works and conclude in Section 6.

## 2   Notation and Preliminaries

In this section, we introduce our setting, which closely follows [14]. We denote the input features by $x \in \mathcal{X}$ where $\mathcal{X}$ is the input domain. The particular structure of $\mathcal{X}$ is not of the key importance for this study. The output variables, that are in the center of our analysis, will be denoted by $\hat{y} \in \hat{\mathcal{Y}}$ with $\hat{\mathcal{Y}}$ being the set of possible predictions or the output domain.[5] In such settings as structured prediction or ranking, the predictions are very high-dimensional and with some structure that is useful to model explicitly (for example, a sequence, permutation or image).

The central object of our study is the *loss function* $L(\hat{y}, y) \geq 0$ that represents the cost of making the prediction $\hat{y} \in \hat{\mathcal{Y}}$ when the ground-truth label is $y \in \mathcal{Y}$. Note that in some applications of interest the sets $\hat{\mathcal{Y}}$ and $\mathcal{Y}$ are different. For example, in ranking with the mean average precision (mAP) loss function (see Section 5.2 and, e.g., [18] for the details), the set $\hat{\mathcal{Y}}$ consists of all the permutations of the items (to represent the ranking itself), but the set $\mathcal{Y}$ consists of all the subsets of items (to represent the set of relevant items, which is the ground-truth annotation in this setting). In this paper, we only study the case when both $\hat{\mathcal{Y}}$ and $\mathcal{Y}$ are finite. We denote the cardinality of $\hat{\mathcal{Y}}$ by $k$, and the cardinality of $\mathcal{Y}$ by $m$. In this case, the loss function can be encoded as a matrix $L$ of size $k \times m$.

In many applications of interest, both quantities $k$ and $m$ are exponentially large in the size of the natural dimension of the input $x$. For example, in the task of sequence labeling, both $k$ and $m$ are equal to the number of all possible sequences of symbols from a finite alphabet. In the task of ranking (the mAP formulation), $k$ is equal to the number of permutations of items and $m$ is equal to the number of item subsets.

Following usual practices, we work with the prediction model defined by a (learned) vector-valued *score function* $\mathfrak{f} : \mathcal{X} \to \mathbb{R}^k$, which defines a scalar score $\mathfrak{f}_{\hat{\boldsymbol{y}}}(\boldsymbol{x})$ for each possible output $\hat{\boldsymbol{y}} \in \hat{\mathcal{Y}}$. The final prediction is then chosen as an output configuration with the maximal score:

$$\text{pred}(\mathfrak{f}(\boldsymbol{x})) := \underset{\hat{\boldsymbol{y}} \in \hat{\mathcal{Y}}}{\text{argmax}}\, \mathfrak{f}_{\hat{\boldsymbol{y}}}(\boldsymbol{x}). \tag{1}$$

If the maximal score is given by multiple outputs $\hat{\boldsymbol{y}}$ (so-called *ties*), the predictor follows a simple deterministic tie-breaking rule and picks the output appearing first in some predefined ordering on $\hat{\mathcal{Y}}$.

In this setup, learning consists in finding a *score function* $\mathfrak{f}$ for which the predictor gives the smallest expected loss with features $\boldsymbol{x}$ and labels $\boldsymbol{y}$ coming from an unknown data-generating distribution $\mathcal{D}$:

$$\mathcal{R}_L(\mathfrak{f}) := \mathbf{E}_{(\boldsymbol{x},\boldsymbol{y}) \sim \mathcal{D}}\, L\big(\text{pred}(\mathfrak{f}(\boldsymbol{x})), \boldsymbol{y}\big). \tag{2}$$

The quantity $\mathcal{R}_L(\mathfrak{f})$ is usually referred to as the actual (or population) *risk* based on the loss $L$. Minimizing the actual risk directly is usually difficult (because of non-convexity and non-continuity of the predictor (1)). The standard approach is to substitute (2) with another objective, a *surrogate risk* (or the $\Phi$-risk), which is easier for optimization (in this paper, we only consider convex surrogates):

$$\mathcal{R}_\Phi(\mathfrak{f}) := \mathbf{E}_{(\boldsymbol{x},\boldsymbol{y}) \sim \mathcal{D}}\, \Phi(\mathfrak{f}(\boldsymbol{x}), \boldsymbol{y}), \tag{3}$$

where we will refer to the function $\Phi : \mathbb{R}^k \times \mathcal{Y} \to \mathbb{R}$ as the *surrogate loss*. To make the minimization of (3) well-defined, we will always assume the surrogate loss $\Phi$ to be bounded from below and continuous.

The surrogate loss should be chosen in such a way that the minimization of (3) also leads to the minimization of (2), i.e., to the solution of the original problem. The property of consistency of the surrogate loss is an approach to formalize this intuition, i.e., to guarantee that no matter the data-generating distribution, minimizing (3) w.r.t. $\mathfrak{f}$ implies minimizing (2) w.r.t. $\mathfrak{f}$ as well (both of these are possible only in the limit of infinite data and computational budget). Osokin et al. [14] quantified what happens if the surrogate risk is minimized approximately by translating the optimization error of (3) to the optimization error of (2). The main goal of this paper is to generalize this analysis to the cases when the surrogate is not consistent and to show that there can be trade-offs between the minimum value of the actual risk that can be achieved by minimizing an inconsistent surrogate and the speed with which this minimum can be achieved.

## 3  Calibration Functions and Consistency

In this section, we review the approach of Osokin et al. [14] for studying consistency in the context of structured prediction. The first part of the analysis establishes the connection between the minimization of the actual risk $\mathcal{R}_L$ (2) and the surrogate risk $\mathcal{R}_\Phi$ (3) via the so-called *calibration function* (see Definition 1 [14, and references therein]). This step is usually called *non-parametric* (or pointwise) because it does not explicitly model the dependency of the scores $\boldsymbol{f} := \mathfrak{f}(\boldsymbol{x})$ on the input variables $\boldsymbol{x}$. The second part of the analysis establishes the connection with an optimization algorithm allowing to make a statement about how many iterations would be enough to find a predictor that is (in expectation) within $\varepsilon$ of the global minimum of the actual risk $\mathcal{R}_L$.

**Non-parametric analysis.** The standard non-parametric setting considers all measurable score functions $\mathfrak{f}$ to effectively ignore the dependency on the features $\boldsymbol{x}$. As noted by [14], it is beneficial to consider a restricted set of the score functions $\mathfrak{F}_{\mathcal{F}}$ that consists of all vector-valued Borel measurable functions $\mathfrak{f} : \mathcal{X} \to \mathcal{F}$ where $\mathcal{F} \subseteq \mathbb{R}^k$ is a subspace of allowed score vectors. Compatibility of the subspace $\mathcal{F}$ and the loss function $L$ will be a crucial point of this paper. Note that the analysis is still non-parametric because the dependence on $\boldsymbol{x}$ is not explicitly modeled.

Within the analysis, we will use the *conditional* actual and surrogate risks defined as the expectations of the corresponding losses w.r.t. a categorical distribution $\boldsymbol{q}$ on the set of annotations $\mathcal{Y}$, $m := |\mathcal{Y}|$:

$$\ell(\boldsymbol{f}, \boldsymbol{q}) := \sum\nolimits_{\boldsymbol{y}=1}^m q_{\boldsymbol{y}} L(\text{pred}(\boldsymbol{f}), \boldsymbol{y}), \quad \phi(\boldsymbol{f}, \boldsymbol{q}) := \sum\nolimits_{\boldsymbol{y}=1}^m q_{\boldsymbol{y}} \Phi(\boldsymbol{f}, \boldsymbol{y}). \tag{4}$$

Hereinafter, we represent an $m$-dimensional categorical distribution $\boldsymbol{q}$ as a point in the probability simplex $\Delta_m$ and use the symbol $q_{\boldsymbol{y}}$ to denote the probability of the $\boldsymbol{y}$-th outcome. Using this notation, we can rewrite the risk $\mathcal{R}_L$ and surrogate risk $\mathcal{R}_\Phi$ as

$$\mathcal{R}_L(\mathfrak{f}) = \mathbf{E}_{\boldsymbol{x} \sim \mathcal{D}_\mathcal{X}}\, \ell(\mathfrak{f}(\boldsymbol{x}), \mathbf{P}_\mathcal{D}(\cdot \mid \boldsymbol{x})), \quad \mathcal{R}_\Phi(\mathfrak{f}) = \mathbf{E}_{\boldsymbol{x} \sim \mathcal{D}_\mathcal{X}}\, \phi(\mathfrak{f}(\boldsymbol{x}), \mathbf{P}_\mathcal{D}(\cdot \mid \boldsymbol{x})), \tag{5}$$

where $\mathcal{D}_{\mathcal{X}}$ is the marginal distribution of $\boldsymbol{x}$ and $\mathbf{P}_{\mathcal{D}}(\cdot \mid \boldsymbol{x})$ denotes the conditional distribution of $\boldsymbol{y}$ given $\boldsymbol{x}$ (both defined for the joint data-generating distribution $\mathcal{D}$).

For each score vector $\boldsymbol{f} \in \mathcal{F}$ and a distribution $\boldsymbol{q} \in \Delta_m$ over ground-truth labels, we now define the *excess* actual and surrogate risks

$$\delta\phi(\boldsymbol{f}, \boldsymbol{q}) = \phi(\boldsymbol{f}, \boldsymbol{q}) - \inf_{\hat{\boldsymbol{f}} \in \mathcal{F}} \phi(\hat{\boldsymbol{f}}, \boldsymbol{q}), \quad \delta\ell(\boldsymbol{f}, \boldsymbol{q}) = \ell(\boldsymbol{f}, \boldsymbol{q}) - \inf_{\hat{\boldsymbol{f}} \in \mathbb{R}^k} \ell(\hat{\boldsymbol{f}}, \boldsymbol{q}), \qquad (6)$$

which show how close the current conditional actual and surrogate risks are to the corresponding minimal achievable conditional risks (depending only on the distribution $\boldsymbol{q}$). Note that the two infima in (6) are defined w.r.t. different sets of score vectors. For the surrogate risk, the infimum is taken w.r.t. the set of allowed scores $\mathcal{F}$ capturing only the scores obtainable by the learning process. For the actual risk, the infimum is taken w.r.t. the set of all possible scores $\mathbb{R}^k$ including score vectors that cannot be learned. This distinction is important when analyzing inconsistent surrogates and allows to characterize the *approximation error* of the selected function class.[6]

We are now ready to define the *calibration function*, which is the final object of the non-parametric part of the analysis. Calibration functions directly show how well one needs to minimize the surrogate risk to guarantee that the excess of the actual risk is smaller than $\varepsilon$.

**Definition 1** (Calibration function, [14]). *For a task loss $L$, a surrogate loss $\Phi$, a set of feasible scores $\mathcal{F}$, the* calibration function $H_{\Phi,L,\mathcal{F}}(\varepsilon)$ *is defined as:*

$$H_{\Phi,L,\mathcal{F}}(\varepsilon) := \inf_{\boldsymbol{f} \in \mathcal{F}, \ \boldsymbol{q} \in \Delta_m} \delta\phi(\boldsymbol{f}, \boldsymbol{q}) \qquad (7)$$

$$\text{s.t.} \quad \delta\ell(\boldsymbol{f}, \boldsymbol{q}) \geq \varepsilon, \qquad (8)$$

*where $\varepsilon \geq 0$ is the target accuracy. We set $H_{\Phi,L,\mathcal{F}}(\varepsilon)$ to $+\infty$ when the feasible set is empty.*

By construction, $H_{\Phi,L,\mathcal{F}}$ is non-decreasing on $[0, +\infty)$, $H_{\Phi,L,\mathcal{F}}(\varepsilon) \geq 0$ and $H_{\Phi,L,\mathcal{F}}(0) = 0$. The calibration function also provides the so-called *excess risk bound*

$$H_{\Phi,L,\mathcal{F}}(\delta\ell(\boldsymbol{f}, \boldsymbol{q})) \leq \delta\phi(\boldsymbol{f}, \boldsymbol{q}), \ \forall \boldsymbol{f} \in \mathcal{F}, \ \forall \boldsymbol{q} \in \Delta_m, \qquad (9)$$

which implies the formal connection between the surrogate and task risks [14, Theorem 2].

The calibration function can fully characterize consistency of the setting defined by the surrogate loss, the subspace of scores and the task loss. The maximal value of $\varepsilon$ at which the calibration function $H_{\Phi,L,\mathcal{F}}(\varepsilon)$ equals zero shows the best accuracy on the actual loss that can be obtained [14, Theorem 6]. The notion of level-$\eta$ consistency captures this effect.

**Definition 2** (level-$\eta$ consistency, [14]). *A surrogate loss $\Phi$ is consistent up to level $\eta \geq 0$ w.r.t. a task loss $L$ and a set of scores $\mathcal{F}$ if and only if the calibration function satisfies $H_{\Phi,L,\mathcal{F}}(\varepsilon) > 0$ for all $\varepsilon > \eta$ and there exists $\hat{\varepsilon} > \eta$ such that $H_{\Phi,L,\mathcal{F}}(\hat{\varepsilon})$ is finite.*

The case of level-0 consistency corresponds to the classical consistent surrogate and Fisher consistency. When $\eta > 0$, the surrogate is not consistent, meaning that the actual risk cannot be minimized globally. However, Osokin et al. [14, Appendix E.4] give an example where even though constructing a consistent setting is possible (by the choice of the score subspace $\mathcal{F}$), it might still be beneficial to use only a level-$\eta$ consistent setting because of the exponentially faster growth of the calibration function. The main contribution of this paper is a lower bound on the calibration function (Theorem 3), which is non-zero for $\eta > 0$ and thus can be used to obtain convergence rates in inconsistent settings.

**Optimization and learning guarantees; normalizing the calibration function.** Osokin et al. [14] note that the scale of the calibration function is not defined, i.e., if one multiplies the surrogate loss by some positive constant, the calibration function is multiplied by the same constant as well. One way to define a "natural normalization" is to use a scale-invariant convergence rate of a stochastic optimization algorithm. Osokin et al. [14, Section 3.3] applied the classical online ASGD [13] (under the well-specification assumption) and got the sample complexity (and the convergence rate of ASGD at the same time) result saying that $N^*$ steps of ASGD are sufficient to get $\varepsilon$-accuracy on the *task loss* (in expectation), where $N^*$ is computed as follows:

$$N^* := \frac{4D^2 M^2}{\check{H}^2_{\Phi,L,\mathcal{F}}(\varepsilon)}. \qquad (10)$$

Here the quantity $N^*$ depends on a convex lower bound $\breve{H}_{\Phi,L,\mathcal{F}}(\varepsilon)$ on the calibration function $H_{\Phi,L,\mathcal{F}}(\varepsilon)$ and the constants $D$, $M$, which appear in the convergence rate of ASGD: $D$ is an upper bound on the norm of an optimal solution and $M^2$ is an upper bound on the expected square norm of the stochastic gradient. Osokin et al. [14] show how to bound the constant $DM$ for a very specific quadratic surrogate defined below (see Section 3.1).

## 3.1 Bounds for the Quadratic Surrogate

The major complication in applying and interpreting the theoretical results presented in Section 3 is the complexity of computing the calibration function. Osokin et al. [14] analyzed the calibration function only for the quadratic surrogate

$$\Phi_{\text{quad}}(\boldsymbol{f}, \hat{\boldsymbol{y}}) := \tfrac{1}{2k}\|\boldsymbol{f} + L(:, \boldsymbol{y})\|_2^2 = \tfrac{1}{2k}\sum\nolimits_{\hat{\boldsymbol{y}} \in \hat{\mathcal{Y}}}(f_{\hat{\boldsymbol{y}}}^2 + 2f_{\hat{\boldsymbol{y}}}L(\hat{\boldsymbol{y}}, \boldsymbol{y}) + L(\hat{\boldsymbol{y}}, \boldsymbol{y})^2). \qquad (11)$$

For any task loss $L$, this surrogate is consistent whenever the subspace of allowed scores is rich enough, i.e., the subspace of scores $\mathcal{F}$ fully contains $\text{span}(L)$. To connect with optimization, we assume a parametrization of the subspace $\mathcal{F}$ as a span of the columns of some matrix $F$, i.e., $\mathcal{F} = \text{span}(F) = \{\boldsymbol{f} = F\boldsymbol{\theta} \mid \boldsymbol{\theta} \in \mathbb{R}^r\}$.[7] In the interesting settings, the dimension $r$ is much smaller than both $k$ and $m$. Note that to compute the gradient of the objective (11) w.r.t. the parameters $\boldsymbol{\theta}$, one needs to compute matrix products $F^\mathsf{T}F \in \mathbb{R}^{r \times r}$ and $F^\mathsf{T}L(:, \boldsymbol{y}) \in \mathbb{R}^r$, which are usually both of feasible sizes, but require exponentially big sum ($k$ summands) inside. Computing these quantities can be seen as some form of inference required to run the learning process.

Osokin et al. [14] proved a lower bound on the calibration functions for the quadratic surrogate (11) [14, Theorem 7], which we now present to contrast our result presented in Section 4. When the subspace of scores $\mathcal{F}$ contains $\text{span}(L)$, $\text{span}(L) \subseteq \mathcal{F}$, implying that the setting is consistent, the calibration function is bounded from below by $\min_{i \neq j} \frac{\varepsilon^2}{2k\|P_{\mathcal{F}}\Delta_{ij}\|_2^2}$, where $P_{\mathcal{F}}$ is the orthogonal projection on the subspace $\mathcal{F}$ and $\Delta_{ij} := \mathbf{e}_i - \mathbf{e}_j \in \mathbb{R}^k$ with $\mathbf{e}_c$ being the $c$-th basis vector of the standard basis in $\mathbb{R}^k$. They also showed that for some very structured losses (Hamming and block 0-1 losses), the quantity $k\|P_{\mathcal{F}}\Delta_{ij}\|_2^2$ is not exponentially large and thus the calibration function suggests that efficient learning is possible. One interesting case not studied by Osokin et al. [14] is the situation where the subspace of scores $\mathcal{F}$ does not fully contain the subspace $\text{span}(L)$. In this case, the surrogate might not be consistent but still lead to effective and efficient practical algorithms.

**Normalizing the calibration function.** The normalization constant $DM$ appearing in (10) can also be computed for the quadratic surrogate (11) under the assumption of well-specification (see [14, Appendix F] for details). In particular, we have $DM = L_{\max}^2\xi(\kappa(F)\sqrt{r}RQ_{\max})$, $\xi(z) = z^2 + z$, where $L_{\max}$ denotes the maximal value of all elements in $L$, $\kappa(F)$ is the condition number of the matrix $F$ and $r$ in an upper bound on the rank of $\mathcal{F}$. The constants $R$ and $Q_{\max}$ come from the kernel ASGD setup and, importantly, depend only on the data distribution, but not on the loss $L$ or score matrix $F$. Note that for a given subspace $\mathcal{F}$, the choice of matrix $F$ is arbitrary and it can always be chosen as an orthonormal basis of $\mathcal{F}$ giving a $\kappa(F)$ of one. However, such $F$ can lead to inefficient prediction (1), which makes the whole framework less appealing. Another important observation coming from the value of $DM$ is the justification of the $\frac{1}{k}$ scaling in front of the surrogate (11).

## 4 Calibration Function for Inconsistent Surrogates

Our main result generalizes the Theorem 7 of [14] to the case of inconsistent surrogates (the key difference consists in the absence of the assumption $\text{span}(L) \subseteq \mathcal{F}$).

**Theorem 3** (Lower bound on the calibration function $H_{\Phi_{\text{quad}},L,\mathcal{F}}(\varepsilon)$)**.** *For any task loss $L$, its quadratic surrogate $\Phi_{\text{quad}}$, and a score subspace $\mathcal{F}$, the calibration function is bounded from below:*

$$H_{\Phi_{\text{quad}},L,\mathcal{F}}(\varepsilon) \geq \min_{i \neq j}\max_{v \geq 0}\frac{(\varepsilon v - \xi_{ij}(v))_+^2}{2k\|P_{\mathcal{F}}\Delta_{ij}\|_2^2}, \quad \text{where} \quad \xi_{ij}(v) := \big\| L^\mathsf{T}(v\mathbf{I}_k - P_{\mathcal{F}})\Delta_{ij} \big\|_\infty, \qquad (12)$$

*where $P_{\mathcal{F}}$ is the orthogonal projection on the subspace $\mathcal{F}$, $(x)_+^2 := [x > 0]x^2$ is the truncation of the parabola to its right branch and $\Delta_{ij} := \mathbf{e}_i - \mathbf{e}_j \in \mathbb{R}^k$ with $\mathbf{e}_c \in \mathbb{R}^k$ being the $c$-th column of the*

*identity matrix* $\mathbf{I}_k$. *By convention, if both numerator and denominator of* (12) *equal zero the whole bound equals zero. If only the denominator equals zero then the whole bound equals infinity (the particular pair of $i$ and $j$ is effectively not considered).*

The proof of Theorem 3 starts with using the idea of [14] to compute the calibration function by solving a collection of convex quadratic programs (QPs). Then we diverge from the proof of [14] (because it leads to a non-informative bound in inconsistent settings). For each of the formulated QPs, we construct a dual by using the approach of Dorn [10]. The dual of Dorn is convenient for our needs because it does not require inverting the matrix defining the quadratic terms (compared to the standard Lagrangian dual). The complete proof is given in Appendix B.

**Remark 4.** *The numerator of the bound* (12) *explicitly specifies the point at which the bound becomes non-zero, implying level-$\eta$ consistency with $\eta = \frac{\xi_{ij}(v)}{v}$ for the values of $i$, $j$, $v$ that are active for a particular $\varepsilon$. The quantity $\frac{v^2}{2k\|P_\mathcal{F}\Delta_{ij}\|_2^2}$ bounds the weight of the $\varepsilon^2$ term in the calibration function after it leaves zero. Moving the quantity $v$ defines the trade-off between the slope, which is related to the convergence speed of the algorithm, and the value of $\eta$ defining the best achievable accuracy.*

**Remark 5.** *If we have conditions of Theorem 7 of [14] satisfied, i.e., $\mathrm{span}(L) \subseteq \mathcal{F}$, then the vector $L^\mathsf{T}(\mathbf{I}_k - P_\mathcal{F})\Delta_{ij}$ equals zero and $\xi_{ij}(v)$ becomes $|v-1|\,\|L^\mathsf{T}\Delta_{ij}\|_\infty$, which equals zero when $v = 1$. It might seem that having $v > 1$ can potentially give us a tighter lower bound than Theorem 7 [14] even in consistent cases. However, the quantity $\|L^\mathsf{T}\Delta_{ij}\|_\infty$ upper bounds the maximal possible (w.r.t. the conditional distribution $\mathbf{P}_\mathcal{D}(\cdot \mid \boldsymbol{x})$) value of the excess task loss for a fixed pair $i$, $j$ leading to the identity $v\varepsilon - |v-1|\,\|L^\mathsf{T}\Delta_{ij}\|_\infty = \|L^\mathsf{T}\Delta_{ij}\|_\infty$ for $\varepsilon = \|L^\mathsf{T}\Delta_{ij}\|_\infty$ and $v \geq 1$. Together with the convexity of the function $(x)_+^2$, this implies that the best possible value of $v$ in consistent settings equals one.*

**Remark 6.** *Setting $v$ in* (12) *to any non-negative constant gives a valid lower bound. In particular, setting $v$ to 1 (while potentially making the bound less tight) highlights the separation between the weight of the quadratic term and the best achievable accuracy $\eta$. The bound now reads as follows:*

$$H_{\Phi_{\mathrm{quad}},L,\mathcal{F}}(\varepsilon) \geq \min_{i \neq j} \frac{(\varepsilon - \xi_{ij})_+^2}{2k\|P_\mathcal{F}\Delta_{ij}\|_2^2}, \quad \text{where} \quad \xi_{ij} := \big\| L^\mathsf{T}(\mathbf{I}_k - P_\mathcal{F})\Delta_{ij} \big\|_\infty. \tag{13}$$

*Note that the weight of the $\varepsilon^2$ term now equals the corresponding coefficient of the bound of Theorem 7 [14]. Notably, this weight depends only on the score subspace $\mathcal{F}$, but not on the loss $L$.*

## 5 Bounds for Particular Losses

### 5.1 Multi-Class Classification with the Tree-Structured Loss

As an illustration of the obtained lower bound (12), we consider the task of multi-class classification and the *tree-structured loss*, which is defined for a weighted tree built on labels (such trees on labels often appear in settings with large number of labels, e.g., extreme classification [6]). Leaves in the tree correspond to the class labels $\hat{y} \in \hat{\mathcal{Y}} = \mathcal{Y}$ and the loss function is defined as the length of the path $\rho$ between the leaves, i.e., $L_{\mathrm{tree}}(y, \hat{y}) := \rho(y, \hat{y})$. To compute the lower bound exactly, we assume that the number of children $d_s$ and the weights of the edges connecting a node with its children $\frac{\alpha_s}{2}$ are equal for all the nodes of the same depth level $s = 0, \ldots, D-1$ (see Figure 2 in Appendix C for an example of such a tree) and that $\sum_{s=0}^{D-1} \alpha_s = 1$, which normalizes $L_{\max}$ to one.

To define the score matrix $\mathcal{F}_{\mathrm{tree},s_0}$, we set the *consistency depth* $s_0 \in \{1, \ldots, D\}$ and restrict the scores $\boldsymbol{f}$ to be equal for the groups (blocks) of leaves that have the same ancestor on the level $s_0$. Let $B(i)$ be the set of leaves that have the same ancestor as a leaf $i$ at the depth $s_0$. With this notation, we have $\mathcal{F}_{\mathrm{tree},s_0} = \mathrm{span}\{\sum_{i \in B(j)} \mathbf{e}_i \mid j = 1, \ldots, k\}$. Theorem 3 gives us the bound (see Appendix C):

$$H_{\Phi_{\mathrm{quad}},L_{\mathrm{tree}},\mathcal{F}_{\mathrm{tree},s_0}}(\varepsilon) \geq [\varepsilon > \eta_{s_0}] \frac{(\eta_{s_0} - \bar{\rho}_{s_0} + \alpha_{s_0-1})^2}{(\frac{\eta_{s_0}}{2} + \alpha_{s_0-1})^2} \frac{(\varepsilon - \frac{\eta_{s_0}}{2})_+^2}{4b_{s_0}}, \tag{14}$$

where $b_{s_0}$, $\bar{\rho}_{s_0} := \frac{1}{|B(j)|} \sum_{i \in B(j)} \rho(i,j) = \sum_{s=s_0}^{D-1} \alpha_s \frac{(\prod_{s'=s_0}^{s} d_{s'}) - 1}{\prod_{s'=s_0}^{s} d_{s'}}$ and $\eta_{s_0} := \max_{i \in B(j)} \rho(i,j) = \sum_{s=s_0}^{D-1} \alpha_s$ are the number of blocks, the average and maximal distance within a block, respectively.

Now we discuss the behavior of the bound (14) when changing the truncation level $s_0$. With the growth of $s_0$, the level of consistency $\eta_{s_0}$ goes to $0$ indicating that more labels can be distinguished. At the same time, we have $\frac{\eta_{s_0}}{2} \leq \bar{\rho}_{s_0}$ for the trees we consider and thus the coefficient in front of the

$\varepsilon^2$ term can be bounded from above by $\frac{1}{4b_{s_0}}$, which means that the lower bound on the calibration function decreases at an exponential rate with the growth of $s_0$. These arguments show the trade-off between the level of consistency and the coefficient of $\varepsilon^2$ in the calibration function.

Finally, note that the mixture of 0-1 and block 0-1 losses considered in [14, Appendix E.4] is an instance of the tree-structured loss with $D = 2$. Their bound [14, Proposition 17] matches (14) up to the difference in the definition of the calibration function (they do not have the $[\varepsilon > \eta_{s_0}]$ multiplier because they do not consider pairs of labels that fall in the same block).

## 5.2 Mean Average Precision (mAP) Loss for Ranking

The mAP loss, which is a popular way of measuring the quality of ranking, has attracted significant attention from the consistency point of view [4, 5, 18]. In the mAP setting, the ground-truth labels are binary vectors $\boldsymbol{y} \in \mathcal{Y} = \{0, 1\}^r$ that indicate the items relevant for the query (a subset of $r$ items-to-rank) and the prediction consists in producing a permutation of items $\sigma \in \hat{\mathcal{Y}}, \hat{\mathcal{Y}} = \mathcal{S}_r$. The mAP loss is based on averaging the precision at different levels of recall and is defined as follows:

$$L_{\text{mAP}}(\sigma, y) := 1 - \frac{1}{|\boldsymbol{y}|} \sum_{p:y_p=1}^{r} \frac{1}{\sigma(p)} \sum_{q=1}^{\sigma(p)} y_{\sigma^{-1}(q)} = 1 - \sum_{p=1}^{r} \sum_{q=1}^{p} \frac{1}{\max(\sigma(p), \sigma(q))} \frac{y_p y_q}{|\boldsymbol{y}|}, \quad (15)$$

where $\sigma(p)$ is the position of an item $p$ in a permutation $\sigma$, $\sigma^{-1}$ is the inverse permutation and $|\boldsymbol{y}| := \sum_{p=1}^{r} y_p$. The second identity provides a convenient form of writing the mAP loss [18] showing that the loss matrix $L_{\text{mAP}}$ is of rank at most $\frac{1}{2}r(r+1)$.[8] The matrix $F_{\text{mAP}} \in \mathbb{R}^{r! \times \frac{1}{2}r(r+1)}$ such that $(F_{\text{mAP}})_{\sigma, pq} := \frac{1}{\max(\sigma(p), \sigma(q))}$ is a natural candidate to define the score subspace $\mathcal{F}$ to get the consistent setting with the quadratic surrogate (11) (Eq. (15) implies that $\text{span}(L_{\text{mAP}}) = \text{span}(F_{\text{mAP}})$).

However, as noted in Section 6 of [18], although the matrix $F_{\text{mAP}}$ is convenient from the consistency point of view (in the setup of [18]), it leads to the prediction problem $\max_{\sigma \in \mathcal{S}_r} (F_{\text{mAP}} \boldsymbol{\theta})_\sigma$, which is a quadratic assignment problem (QAP), and most QAPs are NP-hard.

To be able to predict efficiently, it would be beneficial to have the matrix $F$ with $r$ columns such that sorting the $r$-dimensional $\boldsymbol{\theta}$ would give the desired permutation. It appears that it is possible to construct such a matrix by selecting a subset of columns of matrix $F_{\text{mAP}}$. We define $F_{\text{sort}} \in \mathbb{R}^{r! \times r}$ by $(F_{\text{sort}})_{\sigma, p} := \frac{1}{\sigma(p)}$. A solution of the prediction problem $\max_{\sigma \in \mathcal{S}_r} (F_{\text{sort}} \boldsymbol{\theta})_\sigma$ is simply a permutation that sorts the elements of $\boldsymbol{\theta} \in \mathbb{R}^r$ in the decreasing order (this statement follows from the fact that we can always increase the score $(F_{\text{sort}} \boldsymbol{\theta})_\sigma = \sum_{p=1}^{r} \frac{\theta_p}{\sigma(p)}$ by swapping a pair of non-aligned items).

Most importantly for our study, the columns of the matrix $F_{\text{sort}}$ are a subset of the columns of the matrix $F_{\text{mAP}}$, which indicates that learning with the convenient matrix $F_{\text{sort}}$ might be sufficient for the mAP loss. In what follows, we study the calibration functions for the loss matrix $L_{\text{mAP}}$ and score matrices $F_{\text{mAP}}$ and $F_{\text{sort}}$. In Figure 1a-b, we plot the calibration functions for both $F_{\text{mAP}}$ and $F_{\text{sort}}$ and the lower bounds given by Theorem 3. All the curves were obtained for $r = 5$ (computing the exact values of the calibration functions is exponential in $r$).

Next, we study the behavior of the lower bound (12) for large values of $r$. In Lemma 13 of Appendix D, we show that the denominator of the bound (12) is not exponential in $r$ (we have $2r! \|P_{\mathcal{F}_{\text{sort}}} \Delta_{\pi\omega}\|_2^2 = O(r)$). We also know that $\|P_{\mathcal{F}_{\text{sort}}} \Delta_{\pi\omega}\|_2^2 \leq \|P_{\mathcal{F}_{\text{mAP}}} \Delta_{\pi\omega}\|_2^2$ (because $\mathcal{F}_{\text{sort}}$ is a subspace of $\mathcal{F}_{\text{mAP}}$), which implies that the calibration function of the consistent setting grows not faster than the one of the inconsistent setting. We can also numerically compute a lower bound on the point $\eta$ until which the calibration function is guaranteed to be zero (for this we simply pick two permutations $\pi$, $\omega$ and a labeling $\boldsymbol{y}$ that delivers large values of $(L_{\text{mAP}}^\mathsf{T}(\mathbf{I}_k - P_{\mathcal{F}_{\text{sort}}}) \Delta_{ij})_{\boldsymbol{y}} \leq \xi_{\pi, \omega}(1)$). Figure 1c shows that the level of inconsistency $\eta$ grows with the growth of $r$, which makes the method less appealing for large-scale settings.

Finally, note that to run the ASGD algorithm for the quadratic surrogate (11), mAP loss and score matrix $F_{\text{sort}}$, we need to efficiently compute $F_{\text{sort}}^\mathsf{T} F_{\text{sort}}$ and $F_{\text{sort}}^\mathsf{T} L_{\text{mAP}}(:, \boldsymbol{y})$. Lemmas 11 and 12 (see Appendix D) provide linear in $r$ time algorithms for doing this. The condition number of $\mathcal{F}_{\text{sort}}$ grows as $\Theta(\log r)$ keeping the sample complexity bound (10) well behaved.

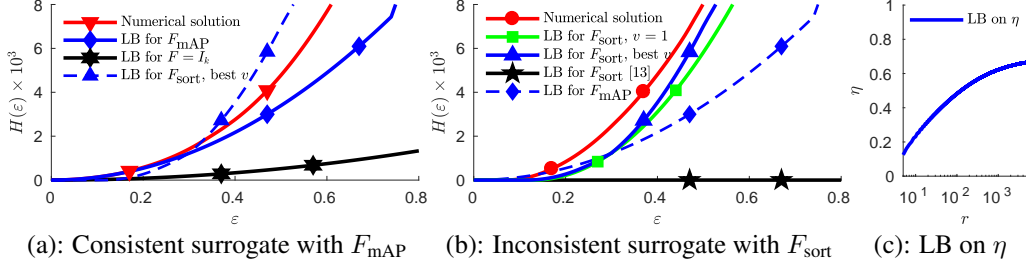

(a): Consistent surrogate with $F_{\text{mAP}}$     (b): Inconsistent surrogate with $F_{\text{sort}}$     (c): LB on $\eta$

Figure 1: **Plot (a)** shows the calibration function $H_{\Phi_{\text{quad}}, L_{\text{mAP}}, \mathcal{F}_{\text{mAP}}}(\varepsilon)$ for $L_{\text{mAP}}$ (red line) obtained numerically. The solid blue line [14, Theorem 7] is its lower bound, LB, and the solid black line is the worst case bound obtained for $F = \mathbf{I}_{r!}$ (which means not constructing an appropriate low-dimension $\mathcal{F}$). Difference between the blue and the black lines is exponential (proportional to $r!$). The dashed blue line illustrates the inconsistent surrogate (note that it is zero for small $\varepsilon > 0$, but then grows faster than the solid blue line – the consistent setting). **Plot (b)** shows the calibration function $H_{\Phi_{\text{quad}}, L_{\text{mAP}}, \mathcal{F}_{\text{sort}}}(\varepsilon)$ (red line) obtained numerically (this setting is level-$\eta$ consistent for $\eta \approx 0.08$). The blue line (Theorem 3) is its lower bound for the optimal value of $v$ and the green line is the bound for $v = 1$ (easier to obtain). The black line shows the zero-valued trivial bound from [14]. The dashed blue line shows $H_{\Phi_{\text{quad}}, L_{\text{mAP}}, \mathcal{F}_{\text{mAP}}}(\varepsilon)$ for the consistent surrogate to compare the two settings. Note that in both plots (a) and (b) the solid blue lines are the lower bounds of the corresponding calibration functions (red lines), but the dashed blue lines are not (shown for comparison purposes). **Plot (c)** shows a lower bound on the point $\eta$ where the exact calibration function $H_{\Phi_{\text{quad}}, L_{\text{mAP}}, \mathcal{F}_{\text{sort}}}(\varepsilon)$ stops being zero, indicating the level of consistency (Definition 2).

## 6  Discussion

**Related works.** Despite a large number of works studying consistency and calibration in the context of machine learning, there have been relatively few attempts to obtain guarantees for inconsistent surrogates. The most popular approach is to study consistency under so-called *low noise* conditions. Such works show that under certain assumptions on the data generating distribution $\mathcal{D}$ (usually these assumptions are on the conditional distribution of labels and are impossible to verify for real data) the surrogate of interest becomes consistent, whereas being inconsistent for general $\mathcal{D}$. Duchi et al. [11] established such a result for the value-regularized linear surrogate for ranking (which resembles the pairwise disagreement, PD, loss). Ramaswamy et al. [18] provided similar results for the mAP and PD losses for ranking and their quadratic surrogate. Similarly to our conclusions, the mAP surrogate of [18] is consistent with $\frac{1}{2}r(r+1)$ parameters learned and only low-noise consistent with $r$ parameters learned. Long & Servedio [12] introduced a notion of realizable consistency w.r.t. a function class (they considered linear predictors), which is consistency w.r.t. the function class assuming the data distribution such that labels depend on features deterministically with this dependency being in the correct function class. Ben-David et al. [3] worked in the agnostic setting for binary classification (no assumptions on the underlying $\mathcal{D}$) and provided guarantees on the error of linear predictors when the margin was bounded by some constant (their work reduces to consistency in the limit case, but is more general).

**Conclusion.** Differently from the previous approaches, we do not put constraints on the data generating distribution, but instead study the connection between the surrogate and task losses by the means of the calibration function (following [14]), which represents the worst-case scenario. For the quadratic surrogate (11), we can bound the calibration function from below in such a way that the bound is non-trivial in inconsistent settings (differently from [14]). Our bound quantifies the level of inconsistency of a setting (defined by the used surrogate loss, task loss and parametrization of the scores) and allows to analyze when learning with inconsistent surrogates can be beneficial. We illustrate the behavior of our bound for two tasks (multi-class classification and ranking) and show examples of conclusions that our approach can give.

**Future work.** It would be interesting to combine our quantitative analysis with the constraints on the data distribution, which might give adaptive calibration functions (in analogy to adaptive convergence rates in convex optimization: for example, SAGA [9] has a linear convergence rate for strongly convex objectives and $1/t$ rate for non-strongly convex ones), and with the recent results of Pillaud-Vivien et al. [16] showing that under some low-noise assumptions even slow convergence of the surrogate objective can imply exponentially fast convergence of the task loss.

**Acknowledgements**

This work was partly supported by Samsung Research, by Samsung Electronics, by the Ministry of Education and Science of the Russian Federation (grant 14.756.31.0001) and by the NSERC Discovery Grant RGPIN-2017-06936.

## Footnotes

[5]The output domain $\hat{\mathcal{Y}}$ itself can depend on the vector of input features $x$ (for example, if $x$ can represent sequences of different lengths and the length of the output sequence has to equal the length of the input), but we will not use this dependency and omit it for brevity.

[6]Note that Osokin et al. [14] define the excess risks by taking both infima w.r.t. the the set of allowed scores $\mathcal{F}$, which is subtly different from us. The results of the two setups are equivalent in the cases of consistent surrogates, which are the main focus of Osokin et al. [14], but can be different in inconsistent cases.

[7]We do a pointwise analysis in this section, so we are not modeling the dependence of $\boldsymbol{\theta}$ on the features $\boldsymbol{x}$. However, in an actual implementation, the vector $\boldsymbol{\theta}$ should be a function of the features $\boldsymbol{x}$ coming from some flexible family such as a RKHS or some neural networks.

[8]Ramaswamy & Agarwal [17, Proposition 21] showed that the rank of $L_{\text{mAP}}$ is a least $\frac{1}{2}r(r+1) - 2$.

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
