[Supplementary Material · struminsky2018_with_supplementary.pdf]

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

}_{\mathrm{mAP}}}\Delta_{\pi\omega}\|_2^2$ (because $\mathcal{F}_{\mathrm{sort}}$ is a subspace of $\mathcal{F}_{\mathrm{mAP}}$), which implies that the calibration function of the consistent setting grows not faster than the one of the inconsistent setting. We can also numerically compute a lower bound on the point $\eta$ until which the calibration function is guaranteed to be zero (for this we simply pick two permutations $\pi$, $\omega$ and a labeling $\boldsymbol{y}$ that delivers large values of $\left(L_{\mathrm{mAP}}^{\mathsf{T}}(\mathbf{I}_k - P_{\mathcal{F}_{\mathrm{sort}}})\Delta_{ij}\right)_{\boldsymbol{y}} \leq \xi_{\pi,\omega}(1)$). Figure 1c shows that the level of inconsistency $\eta$ grows with the growth of $r$, which makes the method less appealing for large-scale settings.

Finally, note that to run the ASGD algorithm for the quadratic surrogate (11), mAP loss and score matrix $F_{\mathrm{sort}}$, we need to efficiently compute $F_{\mathrm{sort}}^{\mathsf{T}}F_{\mathrm{sort}}$ and $F_{\mathrm{sort}}^{\mathsf{T}}L_{\mathrm{

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

[9]Here we show the dual of Dorn [10] for the exact combination of constraints we are using. In the dual formulation, $\boldsymbol{v}$ and $\boldsymbol{u}$ are the extra variables corresponding to the inequality and equality constraints, respectively.

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

# Supplementary Material (Appendix)

# Quantifying Learning Guarantees for Convex but Inconsistent Surrogates

## Outline

## A   Technical Lemmas

In this section, we prove two technical lemmas that are used in the proofs of the main theoretical claims of the paper. These two lemmas are the generalizations of the two corresponding lemmas of [14].

Lemma 7 computes the excess of the weighted surrogate risk $\delta\phi$ for the quadratic loss $\Phi_{\text{quad}}$ (11), which is central to our analysis presented in Section 4. Lemma 7 generalizes Lemma 9 of [14] by removing the assumption of $\text{span}(L) \subseteq \mathcal{F}$. Analogously to Lemma 9 [14], the key property of this result is that the excess $\delta\phi$ is jointly convex w.r.t. the parameters $\boldsymbol{\theta}$ and conditional distribution $\boldsymbol{q}$, which allows further analysis.

Lemma 8 allows to cope with the combinatorial aspect of the calibration function computation. In particular, when the excess of the weighted surrogate risk is convex, Lemma 8 reduces the computation of the calibration function to a set of convex optimization problems, which often can be solved analytically. Note that our Lemma 8 is slightly different from Lemma 10 of Osokin et al. [14] to deal with the difference of the definition of the excess population risk (6).

**Lemma 7.** *Consider the quadratic surrogate $\Phi_{\text{quad}}$ (11) defined for a task loss $L$. Let a subspace of scores $\mathcal{F} \subseteq \mathbb{R}^k$ be parametrized by $\boldsymbol{\theta} \in \mathbb{R}^r$, i.e., $\boldsymbol{f} = F\boldsymbol{\theta} \in \mathcal{F}$ with $F \in \mathbb{R}^{k \times r}$. Then, the excess of the weighted surrogate loss can be expressed as*

$$\delta\phi_{\text{quad}}(F\boldsymbol{\theta}, \boldsymbol{q}) := \phi_{\text{quad}}(F\boldsymbol{\theta}, \boldsymbol{q}) - \inf_{\boldsymbol{\theta}' \in \mathbb{R}^r} \phi_{\text{quad}}(F\boldsymbol{\theta}', \boldsymbol{q}) = \tfrac{1}{2k}\|F\boldsymbol{\theta} + P_{\mathcal{F}}L\boldsymbol{q}\|_2^2,$$

*where $P_{\mathcal{F}} := F(F^{\mathsf{T}}F)^{\dagger}F^{\mathsf{T}}$ is the orthogonal projection on the subspace $\mathcal{F} = \text{span}(F)$.*

*Proof.* The proof is almost identical to the proof of Lemma 9 of [14] generalizing it only in the last equality. By the definition of the quadratic surrogate $\Phi_{\text{quad}}$ (11), we have

$$\phi(\boldsymbol{f}(\boldsymbol{\theta}), \boldsymbol{q}) = \tfrac{1}{2k}(\boldsymbol{\theta}^{\mathsf{T}}F^{\mathsf{T}}F\boldsymbol{\theta} + 2\boldsymbol{\theta}^{\mathsf{T}}F^{\mathsf{T}}L\boldsymbol{q}) + r(\boldsymbol{q}),$$

$$\boldsymbol{\theta}^* := \operatorname{argmin}_{\boldsymbol{\theta}} \phi(\boldsymbol{f}(\boldsymbol{\theta}), \boldsymbol{q}) = -(F^{\mathsf{T}}F)^{\dagger}F^{\mathsf{T}}L\boldsymbol{q},$$

$$\delta\phi(\boldsymbol{f}(\boldsymbol{\theta}), \boldsymbol{q}) = \tfrac{1}{2k}(\boldsymbol{\theta}^{\mathsf{T}}F^{\mathsf{T}}F\boldsymbol{\theta} + 2\boldsymbol{\theta}^{\mathsf{T}}F^{\mathsf{T}}L\boldsymbol{q} + \boldsymbol{q}^{\mathsf{T}}L^{\mathsf{T}}F(F^{\mathsf{T}}F)^{\dagger}F^{\mathsf{T}}L\boldsymbol{q})$$

$$= \tfrac{1}{2k}\|F\boldsymbol{\theta} + P_{\mathcal{F}}L\boldsymbol{q}\|_2^2,$$

where $r(\boldsymbol{q})$ denotes the quantity independent of the parameters $\boldsymbol{\theta}$. Note that if the assumption $\text{span}(L) \subseteq \text{span}(F)$ holds we have $P_{\mathcal{F}}L = L$, which is the statement of Lemma 9 [14]. $\qquad\square$

**Lemma 8.** *For any task loss $L$, a surrogate loss $\Phi$ that is continuous and bounded from below, and a set of scores $\mathcal{F}$, the calibration function can be lower bounded as*

$$H_{\Phi,L,\mathcal{F}}(\varepsilon) \geq \min_{i \neq j} H_{ij}(\varepsilon), \tag{16}$$

*where $H_{ij}$ is defined via minimization of the same objective as (7), but w.r.t. a smaller domain:*

$$H_{ij}(\varepsilon) = \inf_{\boldsymbol{f},\boldsymbol{q}} \delta\phi(\boldsymbol{f},\boldsymbol{q}), \tag{17}$$

$$\text{s.t. } \ell_i(\boldsymbol{q}) \leq \ell_j(\boldsymbol{q}) - \varepsilon,$$
$$\ell_i(\boldsymbol{q}) \leq \ell_c(\boldsymbol{q}), \quad \forall c \in \hat{\mathcal{Y}},$$
$$f_j \geq f_c, \quad \forall c \in \hat{\mathcal{Y}},$$
$$\boldsymbol{f} \in \mathcal{F},$$
$$\boldsymbol{q} \in \Delta_m.$$

*Here $\ell_c(\boldsymbol{q}) := (L\boldsymbol{q})_c$ is the expected loss if predicting label $c$. The index $i$ represents a label with the smallest expected loss while the index $j$ represents a label with the largest score.*

*Proof.* We use the notation $\mathcal{F}_j$ to define the set of score vectors $\boldsymbol{f}$ where the predictor $\mathrm{pred}(\boldsymbol{f})$ takes the value $j$, i.e., $\mathcal{F}_j := \{\boldsymbol{f} \in \mathcal{F} \mid \mathrm{pred}(\boldsymbol{f}) = j\}$. The union of the sets $\mathcal{F}_j$, $j \in \hat{\mathcal{Y}}$, equals the whole set $\mathcal{F}$. Sets $\mathcal{F}_j$ might not contain their boundaries because of the usage of a particular tie-breaking strategy, thus we consider the sets $\overline{\mathcal{F}}_j := \{\boldsymbol{f} \in \mathcal{F} \mid f_j \geq f_c, \forall c \in \hat{\mathcal{Y}}\}$, which are the closures of $\mathcal{F}_j$ if $\mathcal{F}_j$ are not empty. It also might happen that because of a particular tie-breaking strategy a set $\mathcal{F}_j$ is empty, while the corresponding $\overline{\mathcal{F}}_j$ is not.

If $\boldsymbol{f} \in \mathcal{F}_j$, i.e. $j = \mathrm{pred}(\boldsymbol{f})$, then the feasible set of probability vectors $\boldsymbol{q}$ for which a label $i$ is one of the best possible predictions (i.e. $\delta\ell(\boldsymbol{f},\boldsymbol{q}) = \ell_j(\boldsymbol{q}) - \ell_i(\boldsymbol{q}) \geq \varepsilon$) equals

$$\Delta_{m,i,j,\varepsilon} := \{\boldsymbol{q} \in \Delta_m \mid \ell_i(\boldsymbol{q}) \leq \ell_c(\boldsymbol{q}), \forall c \in \hat{\mathcal{Y}}; \ell_j(\boldsymbol{q}) - \ell_i(\boldsymbol{q}) \geq \varepsilon\},$$

because $\inf_{\boldsymbol{f}' \in \mathbb{R}^k} \ell(\boldsymbol{f}',\boldsymbol{q}) = \min_{c \in \hat{\mathcal{Y}}} \ell_c(\boldsymbol{q})$.

The union of the sets $\{\mathcal{F}_j \times \Delta_{m,i,j,\varepsilon}\}_{i,j \in \hat{\mathcal{Y}}, i \neq j}$ exactly equals the feasibility set of the optimization problem (7)-(8) (note that this is not true for the union of the sets $\{\overline{\mathcal{F}}_j \times \Delta_{m,i,j,\varepsilon}\}_{i,j \in \hat{\mathcal{Y}}, i \neq j}$, which can be strictly larger), thus we can rewrite the definition of the calibration function as follows:

$$H_{\Phi,L,\mathcal{F}}(\varepsilon) = \min_{\substack{i,j \in \hat{\mathcal{Y}} \\ i \neq j}} \inf_{\substack{\boldsymbol{f} \in \mathcal{F}_j, \\ \boldsymbol{q} \in \Delta_{m,i,j,\varepsilon}}} \delta\phi(\boldsymbol{f},\boldsymbol{q}) \geq \min_{\substack{i,j \in \hat{\mathcal{Y}} \\ i \neq j}} \inf_{\substack{\boldsymbol{f} \in \overline{\mathcal{F}}_j, \\ \boldsymbol{q} \in \Delta_{m,i,j,\varepsilon}}} \delta\phi(\boldsymbol{f},\boldsymbol{q}) = \min_{i \neq j} H_{ij}(\varepsilon), \tag{18}$$

which finishes the proof. Note that the inequality of (18) can be not tight only if some $\mathcal{F}_j$ is empty, but the corresponding $\overline{\mathcal{F}}_j$ is not (due to continuity of the function $\delta\phi(\boldsymbol{f},\boldsymbol{q})$, which follows from Lemma 27 of [23]). $\square$

## B Proof of Theorem 3

**Theorem 3** (Lower bound on the calibration function $H_{\Phi_{\mathrm{quad}},L,\mathcal{F}}(\varepsilon)$)**.** *For any task loss $L$, its quadratic surrogate $\Phi_{\mathrm{quad}}$, and a score subspace $\mathcal{F}$, the calibration function is bounded from below:*

$$H_{\Phi_{\mathrm{quad}},L,\mathcal{F}}(\varepsilon) \geq \min_{i \neq j} \max_{v \geq 0} \frac{(\varepsilon v - \xi_{ij}(v))_+^2}{2k \|P_{\mathcal{F}}\Delta_{ij}\|_2^2}, \quad \text{where} \quad \xi_{ij}(v) := \left\| L^{\mathsf{T}}(v\mathbf{I}_k - P_{\mathcal{F}})\Delta_{ij} \right\|_\infty, \tag{19}$$

*$P_{\mathcal{F}}$ is the orthogonal projection on the subspace $\mathcal{F}$, $(x)_+^2 := [x > 0]x^2$ is the truncation of the parabola to its right branch and $\Delta_{ij} := \mathbf{e}_i - \mathbf{e}_j \in \mathbb{R}^k$ with $\mathbf{e}_c \in \mathbb{R}^k$ being the $c$-th column of the identity matrix $\mathbf{I}_k$. By convention, if both numerator and denominator of (19) equal zero the whole bound equals zero. If only the denominator equals zero then the whole bound equals infinity (the particular pair of $i$ and $j$ is effectively not considered).*

*Proof.* First, let us assume that the score subspace $\mathcal{F}$ is defined as the column space of a matrix $F \in \mathbb{R}^{k \times r}$, i.e., $\boldsymbol{f}(\boldsymbol{\theta}) = F\boldsymbol{\theta}$. For technical convenience, we can also assume that $F$ is of the full rank, $\mathrm{rank}(F) = r$. Lemma 7 gives us the expression $\delta\phi_{\mathrm{quad}}(F\boldsymbol{\theta},\boldsymbol{q}) = \frac{1}{2k}\|F\boldsymbol{\theta} + P_{\mathcal{F}}L\boldsymbol{q}\|_2^2$ for the excess surrogate, which is jointly convex w.r.t. a conditional probability vector $\boldsymbol{q}$ and parameters $\boldsymbol{\theta}$.

The optimization problem (7)-(8) is non-convex because the constraint (8) on the excess risk depends of the predictor function $\mathrm{pred}(\boldsymbol{f})$, see Eq. (1), containing the $\mathrm{argmax}$ operation. However, if we

constrain the predictor to output label $j$, i.e., $f_j \geq f_c$, $\forall c$, and the label delivering the smallest possible expected loss to be $i$, i.e., $(L\boldsymbol{q})_i \leq (L\boldsymbol{q})_c$, $\forall c$, the problem becomes convex because all the constraints are linear and the objective is convex. Lemma 8 in Appendix A allows to bound the calibration function with the minimal w.r.t. selected labels $i$ and $j$ optimal value of one of the convex problems, i.e., $H_{\Phi_{\text{quad}}, L, \mathcal{F}}(\varepsilon) \geq \min_{i \neq j} H_{ij}(\varepsilon)$, where $H_{ij}(\varepsilon)$ is defined as follows:

$$H_{ij}(\varepsilon) = \min_{\boldsymbol{\theta}, \boldsymbol{q}} \frac{1}{2k} \|F\boldsymbol{\theta} + P_{\mathcal{F}} L\boldsymbol{q}\|_2^2, \tag{20}$$

$$\text{s.t. } (L\boldsymbol{q})_i \leq (L\boldsymbol{q})_j - \varepsilon,$$
$$(L\boldsymbol{q})_i \leq (L\boldsymbol{q})_c, \quad \forall c \in \hat{\mathcal{Y}},$$
$$(F\boldsymbol{\theta})_j \geq (F\boldsymbol{\theta})_c, \quad \forall c \in \hat{\mathcal{Y}},$$
$$\boldsymbol{q} \in \Delta_m.$$

To obtain a lower bound, we relax (20) by removing some of the constraints and arrive at

$$kH_{ij}(\varepsilon) \geq \min_{\boldsymbol{\theta}, \boldsymbol{q}} \frac{1}{2} \|F\boldsymbol{\theta} + P_{\mathcal{F}} L\boldsymbol{q}\|_2^2, \tag{21}$$

$$\text{s.t. } \Delta_{ij}^{\mathsf{T}} L\boldsymbol{q} \leq -\varepsilon, \tag{22}$$

$$\Delta_{ij}^{\mathsf{T}} F\boldsymbol{\theta} \leq 0, \tag{23}$$

$$\mathbf{1}_m^{\mathsf{T}} \boldsymbol{q} = 1, \tag{24}$$

$$q_c \geq 0, \quad c = 1, \ldots, m. \tag{25}$$

where $\Delta_{ij}^{\mathsf{T}} L\boldsymbol{q} = (L\boldsymbol{q})_i - (L\boldsymbol{q})_j$, $\Delta_{ij}^{\mathsf{T}} F\boldsymbol{\theta} = (F\boldsymbol{\theta})_i - (F\boldsymbol{\theta})_j$, and $\Delta_{ij} = \mathbf{e}_i - \mathbf{e}_j \in \mathbb{R}^k$ with $\mathbf{e}_c \in \mathbb{R}^k$ being a vector with 1 at position $c$ and zeros elsewhere. Note that the relaxation defined by the problem (21)-(25) is tighter than the one used in the proof of Theorem 7 [14, Eq. (25)-(27)], because the latter omitted the simplex constraints (24)-(25).

We now explicitly build a dual problem to the QP (21)-(25). If we used the standard Lagrangian approach we would have to invert the matrix defining the objective, which is difficult. Instead we use the dual formulation of Dorn [10, Page 160], which allows to build a dual without inverting any matrices.[9] For the problem (21)-(25), this dual can be written as follows:

$$kH_{ij}(\varepsilon) \geq \max_{\boldsymbol{\theta}, \boldsymbol{q}, v_F \geq 0, v_L \geq 0, u} -\frac{1}{2} \|F\boldsymbol{\theta} + P_{\mathcal{F}} L\boldsymbol{q}\|_2^2 + v_L \varepsilon + u, \tag{26}$$

$$-v_L L^{\mathsf{T}} \Delta_{ij} + u\mathbf{1}_m - L^{\mathsf{T}} P_{\mathcal{F}} L\boldsymbol{q} - L^{\mathsf{T}} F\boldsymbol{\theta} \leq \mathbf{0}_m, \tag{27}$$

$$-v_F F^{\mathsf{T}} \Delta_{ij} - F^{\mathsf{T}} L\boldsymbol{q} - F^{\mathsf{T}} F\boldsymbol{\theta} = \mathbf{0}_r. \tag{28}$$

From the equality (28), we can express $F^{\mathsf{T}} L\boldsymbol{q} = -v_F F^{\mathsf{T}} \Delta_{ij} - F^{\mathsf{T}} F\boldsymbol{\theta}$ and substitute it in the objective (26) and inequality (27). Using the identities $P_{\mathcal{F}} = F(F^{\mathsf{T}} F)^{-1} F^{\mathsf{T}}$ and $P_{\mathcal{F}} F = F$, we can exclude variables $\boldsymbol{\theta}$, $\boldsymbol{q}$ and get a simpler bound. Note that this step leads to a valid lower bound because for any $v_F \geq 0$ there exist feasible values of variables $\boldsymbol{q}$ and $\boldsymbol{\theta}$ (we can take simply $\boldsymbol{q} = \mathbf{0}$, $\boldsymbol{\theta} = -v_F(F^{\mathsf{T}} F)^{-1} F^{\mathsf{T}} \Delta_{ij}$). The new bound depends on the three variables only:

$$kH_{ij}(\varepsilon) \geq \max_{v_F \geq 0, v_L \geq 0, u} -\frac{1}{2} v_F^2 \Delta_{ij}^{\mathsf{T}} P_{\mathcal{F}} \Delta_{ij} + v_L \varepsilon + u, \tag{29}$$

$$-v_L L^{\mathsf{T}} \Delta_{ij} + u\mathbf{1}_m + v_F L^{\mathsf{T}} P_{\mathcal{F}} \Delta_{ij} \leq \mathbf{0}_m. \tag{30}$$

The primal problem

$$\min_{\boldsymbol{q} \geq 0, \boldsymbol{\theta}} \frac{1}{2} \begin{pmatrix} \boldsymbol{q}^{\mathsf{T}} & \boldsymbol{\theta}^{\mathsf{T}} \end{pmatrix} \begin{pmatrix} H_{qq} & H_{q\theta} \\ H_{q\theta}^{\mathsf{T}} & H_{\theta\theta} \end{pmatrix} \begin{pmatrix} \boldsymbol{q} \\ \boldsymbol{\theta} \end{pmatrix},$$
$$\text{s.t. } A_q \boldsymbol{q} + A_\theta \boldsymbol{\theta} \geq \boldsymbol{b}$$
$$C_q \boldsymbol{q} + C_\theta \boldsymbol{\theta} = \boldsymbol{d}$$

The dual problem

$$\max_{\boldsymbol{q}, \boldsymbol{\theta}, \boldsymbol{v} \geq 0, \boldsymbol{u}} -\frac{1}{2} \begin{pmatrix} \boldsymbol{q}^{\mathsf{T}} & \boldsymbol{\theta}^{\mathsf{T}} \end{pmatrix} \begin{pmatrix} H_{qq} & H_{q\theta} \\ H_{q\theta}^{\mathsf{T}} & H_{\theta\theta} \end{pmatrix} \begin{pmatrix} \boldsymbol{q} \\ \boldsymbol{\theta} \end{pmatrix} + \boldsymbol{b}^{\mathsf{T}} \boldsymbol{v} + \boldsymbol{d}^{\mathsf{T}} \boldsymbol{u},$$
$$\text{s.t. } A_q^{\mathsf{T}} \boldsymbol{v} + C_q^{\mathsf{T}} \boldsymbol{u} - H_{qq} \boldsymbol{q} - H_{q\theta} \boldsymbol{\theta} \leq \mathbf{0}$$
$$A_\theta^{\mathsf{T}} \boldsymbol{v} + C_\theta^{\mathsf{T}} \boldsymbol{u} - H_{q\theta}^{\mathsf{T}} \boldsymbol{q} - H_{\theta\theta} \boldsymbol{\theta} = \mathbf{0}$$

Figure 2: *Left:* An example of the tree-structured loss for the task of multi-class classification. *Right:* Illustration of the proof of Lemma 9 (best viewed in color). The thin gray and brown lines show the absolute values of the components of the vector $L_{\text{tree}}^{\mathsf{T}}(v\mathbf{I} - P_{\mathcal{F}_{\text{tree}}})\Delta_{ij}$ as functions of $v$. The bold blue and green lines correspond to the components at which the maximum value is achieved. The bold red line shows the resulting norm $\|L_{\text{tree}}^{\mathsf{T}}(v\mathbf{I} - P_{\mathcal{F}_{\text{tree}}})\Delta_{ij}\|_{\infty}$.

First, consider the case $\Delta_{ij}^{\mathsf{T}} P_{\mathcal{F}} \Delta_{ij} = \|P_{\mathcal{F}}\Delta_{ij}\|_2^2 \neq 0$. Given that $v_F \geq 0$ we can change the variables by introducing $\hat{v}_F := v_F \|P_{\mathcal{F}}\Delta_{ij}\|_2^2$, $v := v_L/v_F$, $\hat{u} := u/v_F$ after which we get

$$kH_{ij}(\varepsilon) \geq \frac{1}{\|P_{\mathcal{F}}\Delta_{ij}\|_2^2} \max_{\hat{v}_F \geq 0, \hat{v}_L \geq 0, \hat{u}} -\tfrac{1}{2}\hat{v}_F^2 + \hat{v}_F(v\varepsilon + \hat{u}), \tag{31}$$

$$-vL^{\mathsf{T}}\Delta_{ij} + \hat{u}\mathbf{1}_m + L^{\mathsf{T}}P_{\mathcal{F}}\Delta_{ij} \leq \mathbf{0}_m. \tag{32}$$

The global minimum of this function w.r.t. the variable $\hat{v}_F$ can be found analytically: if $v\varepsilon + \hat{u} \geq 0$ it equals $\frac{1}{2\|P_{\mathcal{F}}\Delta_{ij}\|_2^2}(v\varepsilon + \hat{u})^2$, and zero otherwise. The constraint (32) on $\hat{u}$ can be substituted with $\hat{u} = -\left\|L^{\mathsf{T}}(v\mathbf{I}_k - P_{\mathcal{F}})\Delta_{ij}\right\|_{\infty} =: -\xi_{ij}(v)$, because we always consider both $H_{ij}(\varepsilon)$ and $H_{ji}(\varepsilon)$ when bounding the calibration function.

Now, consider the boundary case of $\Delta_{ij}^{\mathsf{T}} P_{\mathcal{F}} \Delta_{ij} = \|P_{\mathcal{F}}\Delta_{ij}\|_2^2 = 0$. The problem (29)-(30) becomes $\frac{1}{2}\max_{v \geq 0} v(\varepsilon + \min L^{\mathsf{T}}\Delta_{ij})$ implying that the objective equals 0 if $\varepsilon + \min L^{\mathsf{T}}\Delta_{ij} \leq 0$. Otherwise, the objective equals $+\infty$, which corresponds to the in-feasibility of the constraint (22) of the primal problem. Note that because we always consider both $H_{ij}(\varepsilon)$ and $H_{ji}(\varepsilon)$ when bounding the calibration function we can substitute $v\min(L^{\mathsf{T}}\Delta_{ij})$ with $-\xi_{ij}(v)$. $\qquad\square$

## C  Lower Bound on the Calibration Function for the Tree-Structured Loss

In this section, we compute the lower bound on the calibration function for the tree-structured loss defined in Section 5.1.

**Lemma 9.** *For a particular consistency depth $s_0$ and for the corresponding subspace $\mathcal{F}_{\text{tree},s_0}$, the projection operator $P_{\mathcal{F}_{\text{tree}},s_0}$ at $\Delta_{ij}$ is computed as*

$$P_{\mathcal{F}_{\text{tree}},s_0}\Delta_{ij} = \begin{cases} 0, & i \in B(j), \\ \frac{1}{|B(j)|}\left(\sum_{k \in B(i)} \mathbf{e}_k - \sum_{k \in B(j)} \mathbf{e}_k\right) & i \notin B(j). \end{cases} \tag{33}$$

*The vectors $\xi_{ij}(v)$ are computed as*

$$\xi_{ij}(v) = \begin{cases} v\rho(i,j) & i \in B(j), \\ \max\{|(v-1)\rho(i,j)|, |v(\rho(i,j)-\eta) - (\rho(i,j)-\bar\rho)|\} & i \notin B(j), \end{cases} \tag{34}$$

*for $\eta := \max_{c \in B(i)} \rho(i,c)$ and $\bar\rho := \frac{1}{|B(j)|}\sum_{c \in B(i)} \rho(i,c)$. Finally, the following lower bound of the calibration function for the loss $L_{\text{tree}}$, its quadratic surrogate $\Phi_{\text{quad}}$ and the score subspace $\mathcal{F}_{\text{tree},s_0}$ holds:*

$$H_{\Phi_{\text{quad}},L_{\text{tree}},\mathcal{F}_{\text{tree},s_0}}(\varepsilon) \geq [\varepsilon > \eta]\frac{(\nu - \bar\rho)^2}{(\nu - \frac{\eta}{2})^2}\frac{(\varepsilon - \frac{\eta}{2})_+^2}{4b}, \tag{35}$$

*where $\nu := \min_{c \notin B(i)} \rho(i,c) > \eta$ and $b$ is the number of blocks when the tree is cut at the depth $s_0$.*

*Proof.* For brevity, we shortcut the notation $F_{\text{tree},s_0}$ to $F$, $\mathcal{F}_{\text{tree},s_0}$ to $\mathcal{F}$ and $L_{\text{tree}}$ to $L$. First, we compute the projection operator $P_{\mathcal{F}}\mathbf{e}_i$ and the lower-bound denominator $2k\|P_{\mathcal{F}}\Delta_{ij}\|_2^2$. Recall, that the subspace of allowed scores $\mathcal{F}$ defined as $\text{span}\{\sum_{l\in B(j)}\mathbf{e}_l | j = 1,\ldots,k\}$ is of dimension $b$. The vector $\mathbf{e}_i$ is orthogonal to the $b-1$ different vectors $\sum_{l\in B(j)}\mathbf{e}_l$, $j\notin B(i)$, thus the projection $P_{\mathcal{F}}\mathbf{e}_i$ equals the projection of $\mathbf{e}_i$ on the vector $\sum_{l\in B(i)}\mathbf{e}_l$, which equals $\frac{1}{s}\sum_{l\in B(i)}\mathbf{e}_l$ with $s := B(i) = \frac{k}{b}$. The projection square norm $\|P_{\mathcal{F}}\Delta_{ij}\|_2^2$ equals $\frac{2}{s} = \frac{2b}{k}$ if $i\notin B(j)$ and 0 if $i\in B(j)$.

Next, we compute $\xi_{ij}(v)$ defined as $\|L^{\mathsf{T}}(vI - P_{\mathcal{F}})\Delta_{ij}\|_{\infty}$. By definition of the loss function, the element of the loss matrix $L_{ci} = (L\mathbf{e}_i)_c$ equals the tree distance from the leaf $i$ to the leaf $c$. The projection operator $P_{\mathcal{F}}\mathbf{e}_i$ equals the vector $\frac{1}{s}\sum_{l\in B(i)}\mathbf{e}_l$, therefore $(LP_{\mathcal{F}}\mathbf{e}_i)_c$ is equal to the average tree distance from the elements of the block $B(i)$ to $c$: $\frac{1}{s}\sum_{l\in B(i)}L_{lc}$, which we denote by $\bar{\rho}(i,c)$. Note that the average distance $\bar{\rho}(i,c)$ is equal for all the leaves $c$ that belong to the same block $B(c)$. With this notation, we have the following equality:

$$\|L^{\mathsf{T}}(vI - P_{\mathcal{F}})\Delta_{ij}\|_{\infty} = \max_{c\in\hat{\mathcal{Y}}}|v(\rho(i,c) - \rho(j,c)) - (\bar{\rho}(i,c) - \bar{\rho}(j,c))|. \qquad (36)$$

On the right-hand side, each component is the absolute value of a linear in $v$ function, which equals zero at $v = \frac{\bar{\rho}(i,c)-\bar{\rho}(j,c)}{\rho(i,c)-\rho(j,c)}$ and the absolute value of the slope equals $|\rho(i,c) - \rho(j,c)|$. We consider the cases when the labels $i$ and $j$ are in the same and different blocks separately.

If $i$ and $j$ are in the same block we have that $\bar{\rho}(i,c) = \bar{\rho}(j,c)$ and, by the reverse triangle inequality, $|\rho(i,c) - \rho(j,c)| \leq \rho(i,j)$ with the equality holding for $c = i$ or $c = j$, which implies that $\xi_{ij}(v) = \|(L^{\mathsf{T}}(vI - P_{\mathcal{F}})\Delta_{ij}\|_{\infty} = v\rho(i,j)$ for $i\in B(j)$.

Now, we study the second case where $i$ and $j$ are in different blocks. We first show that

$$|\bar{\rho}(i,c_1) - \bar{\rho}(j,c_1)| \leq |\bar{\rho}(i,c_2) - \bar{\rho}(j,c_2)| \quad \text{if } c_1\notin B(i)\cup B(j) \text{ and } c_2\in B(i)\cup B(j). \qquad (37)$$

This inequality is crucial for the proof and holds due to the restriction on tree weights and node degrees.

In this paragraph, we will show that the left-hand side of the inequality (37) achieves its maximum when $c_1\notin B(i)\cup B(j)$ is in the block closest to $B(j)$ (or, due to the loss symmetries, in the block closest to $B(i)$). If the lowest common ancestor of $i$ and $j$ is not an ancestor of $c_1$ the difference of the average distances equals zero due to equality of the paths from the lowest common ancestor to $i$ and $j$. Otherwise, there exists $c_1\notin B(i)\cup B(j)$ such that the lowest common ancestor of $i$ and $j$ is an ancestor of $c_1$. Then, $\bar{\rho}(j,c_1)$ is minimized and $\bar{\rho}(i,c_1)$ is simultaneously maximized for a component $c_1$ closest to the block $B(j)$. In this case, the left-hand side maximum value equals $\bar{\rho}(i,j) - \min_{c\notin B(j)}\bar{\rho}(j,c)$ because $\bar{\rho}(i,j) = \bar{\rho}(i,c_1)$.

The right-hand side of the inequality (37) is the same for any choice of $c_2\in B(i)\cup B(j)$ and is equal to $\bar{\rho}(i,j) - \bar{\rho}(j,c_2)$ for some $c_2\in B(j)$. Since the average distance within the block is smaller than the average distance to any node outside of the block, i.e., $\min_{c\notin B(j)}\bar{\rho}(j,c) \geq \bar{\rho}(j,c_2)$ for $c_2\in B(j)$, the inequality (37) holds. The same arguments also show that

$$|\rho(i,c_1) - \rho(j,c_1)| \leq |\rho(i,c_2) - \rho(j,c_2)| \quad \text{if } c_1\notin B(i)\cup B(j) \text{ and } c_2\in B(i)\cup B(j). \qquad (38)$$

Recall that in our case the infinity norm in (36) equals the component-wise maximum of the absolute values of the linear functions of $v$. We will show below that for a small enough $v$ the maximum is achieved at the components that have the smallest slope $|\rho(i,c) - \rho(j,c)|$ among the ones with the largest offset $|\bar{\rho}(i,c) - \bar{\rho}(j,c)|$ and from some point for larger values of $v$ the maximum is achieved at the components with the steepest slope (see Figure 2 right for the illustration).

Consider a leaf $c_2\in B(i)$ farthest from the leaf $i$, i.e., $c_2\in\text{argmax}_{c\in B(i)}\rho(i,c)$ (defines the green line in Figure 2 right). The offset $|\bar{\rho}(i,c_2) - \bar{\rho}(j,c_2)|$ is the same for all $c_2\in B(i)$ and, by (37), is larger than the offsets of the components $c_1\notin B(i)\cup B(j)$. The slope $|\rho(i,c_2) - \rho(j,c_2)|$ is the smallest among the components in $B(i)\cup B(j)$. The component $c_2$ of $L^{\mathsf{T}}(v\mathbf{I} - P_{\mathcal{F}})\Delta_{ij}$ equals zero for $v_{c_2}^* := \frac{\bar{\rho}(j,c_2)-\bar{\rho}(i,c_2)}{\rho(j,c_2)-\rho(i,c_2)} = \frac{\rho(i,j)-\bar{\rho}(i,c_2)}{\rho(i,j)-\rho(i,c_2)}$, where $v_{c_2}^* > 1$ by definition of $c_2$, i.e., because $\rho(i,c_2)$ is the maximal distance, which is not smaller than the average distance $\bar{\rho}(i,c_2)$. Finally, for $v \leq 1$ this component has higher values than the values of the components $c\notin B(i)\cup B(j)$. Indeed, the latter

are equal to zero at $v = 1$ and have smaller offset at $v = 0$ (thin brown lines in Figure 2 right for $v < 1$).

The component $i$ of $L^{\mathsf{T}}(v\mathbf{I} - P_{\mathcal{F}})\Delta_{ij}$ has the steepest slope $|\rho(i,j) - \rho(i,i)| = \rho(i,j)$ and the same offset as in the previous paragraph $|\bar{\rho}(i,i) - \bar{\rho}(i,j)| = |\bar{\rho}(i,c_2) - \bar{\rho}(j,c_2)|$ (defines the blue line in Figure 2 right). The component equals zero for $v_i^* := \frac{\bar{\rho}(j,i) - \bar{\rho}(i,i)}{\rho(j,i) - \rho(i,i)} = \frac{\rho(j,i) - \bar{\rho}(i,i)}{\rho(j,i)}$, where $v_i^* \leq 1$. As a result, the component $i$ has higher values than the components $c \notin B(i) \cup B(j)$, since they have smaller slope $|\rho(i,c) - \rho(i,c)|$ (due to the inequality (38)) and equal zero for $v = 1$ (thin brown lines in Figure 2 right for $v > 1$).

Since all the components $c \in B(i) \cup B(j)$ have the same offset, the maximum is achieved either at $c_2$ or at $i$:

$$\|L^{\mathsf{T}}(v\mathbf{I} - P_{\mathcal{F}})\Delta_{ij}\|_\infty = \max\{|v\rho(i,j) - (\rho(i,j) - \bar{\rho})|, |v(\rho(i,j) - \eta) - (\rho(i,j) - \bar{\rho})|\}, \quad (39)$$

where $\eta := \rho(i,c_2)$ is the maximal distance within a block and $\bar{\rho} := \bar{\rho}(i,c_2)$ is the average distance within a block.

Next, we compute $\max_{v \geq 0}(\varepsilon v - \xi_{ij}(v))_+^2$. If $i$ and $j$ are in the same block we have $(\varepsilon v - \xi_{ij}(v))_+^2 = (v(\varepsilon - \rho(i,j)))_+^2$, which equals zero for $\varepsilon \leq \rho(i,j)$ and $+\infty$ otherwise. If $i$ and $j$ are in different blocks we have the maximum value of $(\varepsilon v - \xi_{ij}(v))_+^2$ equal to $+\infty$ when $\varepsilon > \rho(i,j)$. In the case when $\varepsilon \leq \rho(i,j)$, the maximum is achieved at the intersection point $v\rho(i,j) - (\rho(i,j) - \bar{\rho}) = -v(\rho(i,j) - \eta) + (\rho(i,j) - \bar{\rho})$, $v = \frac{2(\rho(i,j) - \bar{\rho})}{2\rho(i,j) - \eta}$. The maximum value is positive if and only if $\varepsilon > \frac{\eta}{2}$, so for $\varepsilon \leq \rho(i,j)$ we obtain

$$\max_{v \geq 0}(\varepsilon v - \xi_{ij}(v))_+^2 = \frac{(\rho(i,j) - \bar{\rho})^2}{(\rho(i,j) - \frac{\eta}{2})^2}(\varepsilon - \frac{\eta}{2})_+^2 \quad (40)$$

and $+\infty$ otherwise.

Finally, to get the actual lower bound on the calibration function, we compute the minimum with respect to all labels $\min_{i \neq j} \max_{v \geq 0}(\varepsilon v - \xi_{ij}(v))_+^2$. When $i$ and $j$ are in the same block, they deliver minimum value 0 for $\varepsilon \leq \rho(i,j)$ and the maximum value of $\rho(i,j)$ within a block equals $\eta$ by definition of $\eta$. For $\varepsilon > \eta$, the minimum is delivered by $i$ and $j$ in different blocks. For the average distance within the block, we have $\bar{\rho} \geq \frac{\eta}{2}$ for the trees with the number of children and the weights of edges equal at the same depth level, therefore the outer minimum w.r.t. $i$ and $j$ is achieved at the smallest distance between two blocks $\nu := \min_{i \notin B(j)} \rho(i,j) > \eta$. As a result, we obtain the bound

$$H_{\Phi_{\mathrm{quad}}, L_{\mathrm{tree}}, \mathcal{F}_{\mathrm{tree}, s_0}}(\varepsilon) \geq [\varepsilon > \eta] \frac{(\nu - \bar{\rho})^2}{(\nu - \frac{\eta}{2})^2} \frac{(\varepsilon - \frac{\eta}{2})_+^2}{4b}, \quad (41)$$

which completes the proof. $\qquad\square$

In the next lemma, we compute the quantities $\eta, \bar{\rho}, \nu$ using the tree weights $\{\frac{1}{2}\alpha_s\}_{s=0}^{D-1}$ to finish the computation of the bound (14) of the main paper.

**Lemma 10.** *For a particular consistency depth $s_0$ and the corresponding subspace $\mathcal{F}_{\mathrm{tree}, s_0}$, the maximum distance within an arbitrary block $\eta_{s_0}$, the minimum distance between a leaf in a block and a leaf outside the block $\nu_{s_0}$ and the average distance within a block $\bar{\rho}_{s_0}$ can be computed as follows:*

$$\eta_{s_0} = \max_{i \in B(j)} \rho(i,j) = \sum_{s=s_0}^{D-1} \alpha_s \quad (42)$$

$$\nu_{s_0} = \min_{i \notin B(j)} \rho(i,j) = \sum_{s=s_0-1}^{D-1} \alpha_s \quad (43)$$

$$\bar{\rho}_{s_0} = \frac{1}{|B(j)|} \sum_{i \in B(j)} \rho(i,j) = \sum_{s=s_0}^{D-1} \alpha_s \frac{(\prod_{s'=s_0}^{s} d_{s'}) - 1}{\prod_{s'=s_0}^{s} d_{s'}}. \quad (44)$$

*Proof.* The expressions for $\eta_{s_0}$ and $\nu_{s_0}$ immediately follow from the definition of the distance $\rho(i,j)$.

To obtain the expression for $\bar{\rho}_{s_0}$, we rewrite the distance $\rho(i, j)$ between leaves $i$ and $j$ in the same block $B(j)$ as the weighted sum of indicators:

$$\rho(i, j) = \sum_{s=s_0}^{D-1} \alpha_s [\text{path from } i \text{ to } j \text{ contains an edge of depth } s]. \tag{45}$$

Then, we fix a leaf $j$ and compute the number of paths from $j$ to the leaves in $B(j)$ that contain an edge of depth $s$. Such paths go through the same node (the ancestor of $j$ at the depth $s$) on the way up from node $j$ and go through one of $\prod_{s'=s_0}^{s} d_{s'} - 1$ possible nodes at the depth $s$ on the way down. From each node of depth $s$, the path can further go to one of $\prod_{s'=s+1}^{D-1} d_{s'}$ leaves on the way down. Therefore, there are $\left(\prod_{s'=s_0}^{s} d_{s'} - 1\right)\left(\prod_{s'=s+1}^{D-1} d_{s'}\right)$ paths that contain an edge of depth $s$.

Next, we rewrite $\sum_{i \in B(j)} \rho(i, j)$ using the indicator notation and compute the sum:

$$\sum_{i \in B(j)} \rho(i, j) = \sum_{s=s_0}^{D-1} \sum_{i \in B(j)} \alpha_s [\text{path from } j \text{ to } i \text{ contains an edge of depth } s] \tag{46}$$

$$= \sum_{s=s_0}^{D-1} \alpha_s \left(\prod_{s'=s_0}^{s} d_{s'} - 1\right)\left(\prod_{s'=s+1}^{D-1} d_{s'}\right). \tag{47}$$

Since the number of leaves in a block is $\prod_{s'=s_0}^{D-1} d_{s'}$, we have

$$\bar{\rho}_{s_0} = \frac{1}{|B(j)|} \sum_{i \in B(j)} \rho(i, j) = \sum_{s=s_0}^{D-1} \alpha_s \frac{(\prod_{s'=s_0}^{s} d_{s'}) - 1}{\prod_{s'=s_0}^{s} d_{s'}}, \tag{48}$$

which finishes the proof. □

Note that for tree-depth $D = 2$ the minimum $\nu_1$ equals $\alpha_0 + \alpha_1 = 1$. As a result, our calibration function lower bound coincides with the exact calibration function from [14].

## D  Derivations for the Mean Average Precision Loss

In this section, we prove several statements about $F_{\text{sort}}$ and $L_{\text{mAP}}$, which are used in Section 5.2.

**Lemma 11.** *The matrix $F_{\text{sort}}^{\mathsf{T}} F_{\text{sort}}$ has the following form:*

$$(F_{\text{sort}}^{\mathsf{T}} F_{\text{sort}})_{pq} = \begin{cases} (r-1)! H_{r,2}, & p = q, \\ (r-2)!(H_{r,1}^2 - H_{r,2}), & p \neq q, \end{cases} \tag{49}$$

*where $H_{n,m} := \sum_{k=1}^{n} \frac{1}{k^m}$ is the generalized harmonic number of order $m$ of $n$. As a result, for distinct permutations $\pi$ and $\omega$, the square norm of the projection is equal to*

$$\|P_{\mathcal{F}_{\text{sort}}} \Delta_{\pi\omega}\|_2^2 = \frac{1}{(r-2)!(rH_{r,2}-H_{r,1}^2)} \sum_{p=1}^{r} \left(\frac{1}{\pi(p)} - \frac{1}{\omega(p)}\right)^2. \tag{50}$$

*The condition number $\kappa(F_{\text{sort}})$ equals $\frac{\sqrt{r-1}H_{r,1}}{\sqrt{rH_{r,2}-H_{r,1}^2}}$.*

*Proof.* By definition, $(F_{\text{sort}}^{\mathsf{T}} F_{\text{sort}})_{pq} = \sum_{\sigma \in S_r} \frac{1}{\sigma(p)\sigma(q)}$. We can rewrite the sum as the sum over the permutations with fixed values $\sigma(p)$ and $\sigma(q)$ and then sum over the fixed values. Therefore, the sum is equal to $(r-1)! H_{r,2}$ when $p = q$ and is equal to $(r-2)!(H_{r,1}^2 - H_{r,2})$ otherwise.

We now have $F_{\text{sort}}^{\mathsf{T}} F_{\text{sort}} = (r-2)!(rH_{r,2} - H_{r,1}^2)\mathbf{I}_r + (r-2)!(H_{r,1}^2 - H_{r,2})\mathbf{1}\mathbf{1}^{\mathsf{T}}$. The Sherman-Woodbury formula for the matrix inversion gives us the sum of the scalar matrix $\frac{1}{(r-2)!(rH_{r,2}-H_{r,1})}\mathbf{I}_r$ and the constant matrix. Since $\mathbf{1}^{\mathsf{T}} F_{\text{sort}}^{\mathsf{T}} \Delta_{\pi\omega} = \sum_{p=1}^{r} \left(\frac{1}{\pi(p)} - \frac{1}{\omega(p)}\right) = 0$, the square norm of the projection equals $\frac{1}{(r-2)!(rH_{r,2}-H_{r,1}^2)} \Delta_{\pi\omega}^{\mathsf{T}} F_{\text{sort}} F_{\text{sort}}^{\mathsf{T}} \Delta_{\pi\omega} = \frac{1}{(r-2)!(rH_{r,2}-H_{r,1}^2)} \sum_{p=1}^{r} \left(\frac{1}{\pi(p)} - \frac{1}{\omega(p)}\right)^2$.

The condition number of $F_{\text{sort}}$ equals the square root of the ratio between the maximal and minimal eigenvalues of $F_{\text{sort}}^{\mathsf{T}}F_{\text{sort}}$. Subtracting $(r-2)!(rH_{r,2}-H_{r,1}^2)\mathbf{I}_r$ from $F_{\text{sort}}^{\mathsf{T}}F_{\text{sort}}$ we get a matrix of rank 1, which means that $r-1$ eigenvalues of $F_{\text{sort}}^{\mathsf{T}}F_{\text{sort}}$ equal $(r-2)!(rH_{r,2}-H_{r,1}^2)$. The remaining eigenvalue corresponds to the eigenvector $\mathbf{1}$ and equals $(r-1)!H_{r,1}^2$. With these eigenvalues we get the condition number $\kappa(F_{\text{sort}})=\dfrac{\sqrt{r-1}H_{r,1}}{\sqrt{rH_{r,2}-H_{r,1}^2}}$. $\qquad\square$

**Lemma 12.** *The matrix $L_{\text{mAP}}^{\mathsf{T}}F_{\text{sort}}$ has the following form:*

$$\left(L_{\text{mAP}}^{\mathsf{T}}F_{\text{sort}}\right)_{y,p} = \begin{cases} \alpha(|\boldsymbol{y}|), & y_p = 1, \\ \beta(|\boldsymbol{y}|), & y_p = 0. \end{cases} \tag{51}$$

*That is, for each ground-truth value $\boldsymbol{y}$ the matrix row components have only two values that depend on the Hamming norm $|\boldsymbol{y}| := \sum_{p=1}^{r} y_p$, specifically:*

$$\alpha(|\boldsymbol{y}|) = \mathfrak{A}_r\left(1-\frac{|\boldsymbol{y}|-1}{r-2}\left(1-\frac{r}{|\boldsymbol{y}|(r-1)}\right)\right)-\mathfrak{B}_r\left(\frac{3}{2}\frac{|\boldsymbol{y}|-1}{r-2}\frac{r-|\boldsymbol{y}|}{|\boldsymbol{y}|}\right)-\mathfrak{C}_r\left(\frac{r-|\boldsymbol{y}|}{|\boldsymbol{y}|(r-1)}\right), \tag{52}$$

$$\beta(|\boldsymbol{y}|) = \mathfrak{A}_r\left(1-\frac{|\boldsymbol{y}|-1}{r-2}\right)-\mathfrak{B}_r\left(1-\frac{3}{2}\frac{|\boldsymbol{y}|-1}{r-2}\right). \tag{53}$$

*Here $\mathfrak{A}_r = (r-1)!H_{r,1}$, $\mathfrak{B}_r = (r-2)!(H_{r,1}^2 - H_{r,2})$, $\mathfrak{C}_r = (r-1)!H_{r,2}$. As a result, for permutations $\pi$ and $\omega$, we obtain*

$$\left(L_{\text{mAP}}^{\mathsf{T}}P_{\mathcal{F}_{\text{sort}}}\Delta_{\pi\omega}\right)_{\boldsymbol{y}} = \gamma(|\boldsymbol{y}|)\left((F_{\text{sort}}\boldsymbol{y})_\pi-(F_{\text{sort}}\boldsymbol{y})_\omega\right), \tag{54}$$

*where $\gamma(p) = \dfrac{\alpha(p)-\beta(p)}{(r-2)!\left(rH_{r,2}-H_{r,1}^2\right)}$.*

*Proof.* For brevity, here we denote $F_{\text{sort}}$ by $F$, $\mathcal{F}_{\text{sort}}$ by $\mathcal{F}$ and $L_{\text{mAP}}$ by $L$. Following the definitions of $L$ and $F$, we explicitly compute the components of $L^{\mathsf{T}}F$:

$$\left(L^{\mathsf{T}}F\right)_{\boldsymbol{y},s} = \sum_{\sigma\in S_r}\left(1-\frac{1}{\boldsymbol{y}}\sum_{p=1}^{r}\sum_{q=1}^{p}\frac{y_{\sigma^{-1}(p)}y_{\sigma^{-1}(q)}}{p}\right)\frac{1}{\sigma(s)}. \tag{55}$$

There are exactly $(r-1)!$ permutations with one fixed element, so we have $\sum_{\sigma\in S_r}\frac{1}{\sigma(s)} = (r-1)!\sum_{p=1}^{r}\frac{1}{p} = (r-1)!H_{r,1} =: \mathfrak{A}_r$. To compute the remaining part of (55), we group the permutation values $\sigma(k) = t$ by each $t = 1,\ldots,r$ and move the sum over permutations inside the bracket:

$$-\frac{1}{\boldsymbol{y}}\sum_{\sigma\in S_r}\sum_{p=1}^{r}\sum_{q=1}^{p}\frac{y_{\sigma^{-1}(p)}y_{\sigma^{-1}(q)}}{p\sigma(s)} = -\frac{1}{\boldsymbol{y}}\sum_{t=1}^{r}\sum_{p=1}^{r}\sum_{q=1}^{p}\sum_{\sigma\in S_r,\sigma(s)=t}\frac{y_{\sigma^{-1}(p)}y_{\sigma^{-1}(q)}}{pt}. \tag{56}$$

Next, we compute the inner sum $\sum_{\sigma\in S_r,\sigma(s)=t}y_{\sigma^{-1}(p)}y_{\sigma^{-1}(q)}$. We rewrite the sum as the sum over inverse permutations:

$$\sum_{\sigma\in S_r,\sigma(s)=t}y_{\sigma^{-1}(p)}y_{\sigma^{-1}(q)} = \sum_{\pi\in S_r,\pi(t)=s}y_{\pi(p)}y_{\pi(q)}. \tag{57}$$

The number of positive terms is different for the two cases of $y_s = 0$ and $y_s = 1$. For $y_s = 0$, using the Iverson brackets the sum can be rewritten as follows:

$$\sum_{\pi\in S_r,\pi(t)=s}y_{\pi(p)}y_{\pi(q)} = [p\neq t]\left([q<p\,\&\,q\neq t]|\boldsymbol{y}|(|\boldsymbol{y}|-1)(r-3)! + [q=p]|\boldsymbol{y}|(r-2)!\right). \tag{58}$$

We then sum the expression over $q = 1,\ldots,p$:

$$\sum_{q=1}^{p}[p\neq t]\left([q<p,q\neq t]|\boldsymbol{y}|(|\boldsymbol{y}|-1)(r-3)! + [q=p]|\boldsymbol{y}|(r-2)!\right) = \tag{59}$$

$$[p\neq t]\left((p-1-[t<p])|\boldsymbol{y}|(|\boldsymbol{y}|-1)(r-3)! + |\boldsymbol{y}|(r-2)!\right). \tag{60}$$

Finally, we multiply the expression by $\frac{-1}{|\boldsymbol{y}|pt}$ and compute the sum over $p$ and $t$:

$$-\frac{1}{|\boldsymbol{y}|}\sum_{t=1}^{r}\sum_{p=1}^{r}\frac{1}{tp}[p \neq t]\left((p-1-[t<p])|\boldsymbol{y}|(|\boldsymbol{y}|-1)(r-3)!+|\boldsymbol{y}|(r-2)!\right)= \tag{61}$$

$$-\mathfrak{A}_r\frac{|\boldsymbol{y}|-1}{r-2}+\mathfrak{B}_r\left(1-\frac{3}{2}\frac{|\boldsymbol{y}|-1}{r-2}\right). \tag{62}$$

Combining with $\sum_{\sigma\in S_r}\frac{1}{\sigma(s)}=\mathfrak{A}_r$ we obtain the desired expression for $\beta(|\boldsymbol{y}|)$.

Now, consider the case of $y_s=1$. Again, we rewrite the sum as

$$\sum_{\pi\in S_r,\pi(t)=s} y_{\pi(p)}y_{\pi(q)} = [p=t]\left([q<p](|\boldsymbol{y}|-1)(r-2)!+[q=p](r-1)!\right) \tag{63}$$

$$+[p\neq t]\left([q<p\ \&\ q=t](|\boldsymbol{y}|-1)(r-2)!\right) \tag{64}$$

$$+[p\neq t]\left([q<p\ \&\ q\neq t](|\boldsymbol{y}|-1)(|\boldsymbol{y}|-2)(r-3)!\right) \tag{65}$$

$$+[p\neq t]\left([q=p](|\boldsymbol{y}|-1)(r-2)!\right), \tag{66}$$

sum it over $q$,

$$\sum_{q=1}^{p}\sum_{\pi\in S_r,\pi(t)=s} y_{\pi(p)}y_{\pi(q)} = [p=t]\left((p-1)(|\boldsymbol{y}|-1)(r-2)!+(r-1)!\right) \tag{67}$$

$$+[p\neq t]\left([t<p](|\boldsymbol{y}|-1)(r-2)!\right) \tag{68}$$

$$+[p\neq t]\left((p-1-[t<q])(|\boldsymbol{y}|-1)(|\boldsymbol{y}|-2)(r-3)!\right) \tag{69}$$

$$+[p\neq t](|\boldsymbol{y}|-1)(r-2)!, \tag{70}$$

and obtain the desired expression by multiplying the latter by $\frac{-1}{|\boldsymbol{y}|pt}$ and summing over $p$ and $t$. The last step of the computation is completely analogous to the case $y_k=0$ and we omit it for brevity.

To compute $\left(L^T P_{\mathcal{F}}\Delta_{\pi\omega}\right)_{\boldsymbol{y}}$, we note that for all permutations $\pi$ and $\omega$ it holds that $\mathbf{1}^\mathsf{T}F^\mathsf{T}\Delta_{\pi\omega}=0$. According to Lemma 11 and the Sherman-Woodbury formula, $(F^\mathsf{T}F)^{-1}$ is the sum of the scalar matrix $\frac{\mathbf{I}_r}{(r-2)!(rH_{r,2}-H_{r,1}^2)}$ and the multiple of the rank one matrix $\mathbf{1}\mathbf{1}^\mathsf{T}$. After the multiplication on $F^\mathsf{T}\Delta_{\pi\omega}$, the second term vanishes, so we get $(F^\mathsf{T}F)^{-1}F^\mathsf{T}\Delta_{\pi\omega}=\frac{\sum_{p=1}^{r}\frac{1}{\pi(p)}-\frac{1}{\omega(p)}}{(r-2)!(rH_{r,2}-H_{r,1}^2)}$. Finally, we rewrite $\left(L^T F\right)_{\boldsymbol{y},:}$ as $(\alpha(|\boldsymbol{y}|)-\beta(|\boldsymbol{y}|))y+\beta(|\boldsymbol{y}|)\mathbf{1}$. By the same argument, after the vector multiplication, the second component vanishes and we get $\left(L^\mathsf{T}F\left(F^\mathsf{T}F\right)^{-1}F\Delta_{\pi\omega}\right)_{\boldsymbol{y}}=\frac{(F\boldsymbol{y})\pi-(F\boldsymbol{y})\omega}{(r-2)!(rH_{r,2}-H_{r,1}^2)}$, which finishes the proof. $\square$

**Lemma 13.** *For the score set $\mathcal{F}_{\text{sort}}$, we have $2(r-1)!\|P_{\mathcal{F}_{\text{sort}}}\Delta_{\pi\omega}\|_2^2=O(r)$. We also have that $\gamma(|\boldsymbol{y}|)$ defined in Lemma 12 with $|\boldsymbol{y}|=\lambda r, \lambda\in(0,1)$ vanishes as $r$ approaches infinity: $\gamma(|\boldsymbol{y}|)=O(\frac{\log^2 r}{r})$. The condition number $\kappa(\mathcal{F}_{\text{sort}})$ grows as $\Theta(\log r)$.*

*Proof.* To derive an asymptotic bound for $\|P_{\mathcal{F}_{\text{sort}}}\Delta_{\pi\omega}\|_2^2$, we elaborate on the sum of squares $\sum_{p=1}^{r}\left(\frac{1}{\pi(p)}-\frac{1}{\omega(p)}\right)^2=2H_{r,2}-2\sum_{p=1}^{r}\frac{1}{\pi(p)\omega(p)}\leq 2H_{r,2}$ and apply the asymptotic bounds for the harmonic numbers $H_{r,1}^2=\Theta(\log^2 r), H_{r,2}=\Theta(1)$:

$$2(r-1)!\|P_{\mathcal{F}_{\text{sort}}}\Delta_{\pi\omega}\|_2^2=O(\frac{(r)!}{(r-2)!r})=O(r) \tag{71}$$

For the second part of the lemma, we rewrite $\alpha(|\boldsymbol{y}|)$ and $\beta(|\boldsymbol{y}|)$:

$$\alpha(|\boldsymbol{y}|)=\mathfrak{A}_r\left(1-\lambda(1-\frac{1}{r})\right)-\mathfrak{B}_r\frac{3}{2}(1-\lambda)-\mathfrak{C}_r\frac{1-\lambda}{r}+o(1) \tag{72}$$

$$\beta(|\boldsymbol{y}|)=\mathfrak{A}_r\left(1-\lambda\right)-\mathfrak{B}_r\left(1-\frac{3}{2}\lambda\right)+o(1) \tag{73}$$

$$\alpha(|\boldsymbol{y}|)-\beta(|\boldsymbol{y}|)=\mathfrak{A}_r\frac{1}{r}-\mathfrak{B}_r\frac{1}{2}-\mathfrak{C}_r\frac{1-\lambda}{r}+o(1) \tag{74}$$

By definition, we have $\mathfrak{A}_r=\Theta((r-1)!\log r), \mathfrak{B}_r=\Theta((r-2)!\log^2 r), \mathfrak{C}_r=\Theta((r-1)!)$, which gives us

$$\gamma(|\boldsymbol{y}|)=\frac{\alpha(|\boldsymbol{y}|)-\beta(|\boldsymbol{y}|)}{(r-2)!(rH_{r,2}-H_{r,1}^2)}=O\left(\frac{(r-2)!\log^2 r}{(r-1)!}\right)=O(\frac{\log^2 r}{r}), \tag{75}$$

what was to be shown.

Finally, the asymptotic bound for the condition number of $\mathcal{F}_{\text{sort}}$ trivially follows from its exact expression in Lemma 11 and the asymptotic bounds for the harmonic numbers. $\square$