[Reviews · NeurIPS 2018]

Reviewer 1



Summary The paper provides a new lower bound in consistency analysis of convex surrogate loss functions. Section 1 provides an introduction which discusses previous work and contributions. The main previous work is Osokin et al, which is frequently referenced in the text. The main contribution is a generalization of the results of Osokin et al to inconsistent surrogates, and a new tighter lower bound. An additional contribution is a study the behavior of the proposed bound for two prediction problems. Section 2 provides definitions of the task loss and surrogate loss, while section 3 defines the calibration function which links them. The concept of level-eta consistency is defined. Level-0 consistency is the same as Fisher consistency, whereas with eta>0 "the surrogate is not consistent meaning that the actual risk cannot be minimized globally." Section 4 provides a new lower bound, with discussion that clarifies how the paper is novel wrt previous work. Section 5 analyzes the bounds in two problem settings: multi-class classification with tree-structured loss and ranking with mean average precision loss. Section 6 provides a discussion and conclusion. Pros: - frequent references to previous work which help clarify the novelty. - new proof technique and lower bound. - results include a comparison with previous work (Figure 1b) in order to show a visual example of how the previous bound is trivial and the proposed bound is tighter. Cons: - no clear discussion of how the analysis may lead to new loss functions or efficient learning algorithms.

Reviewer 2



This paper studied the consistency property of learning via convex surrogates by extending the existing studies to the case of inconsistent surrogates. The main result is Theorem 3 that bounded the calibration function from below. According to my reading of the manuscript, I have the following comments: Positive comments: The deduction of the lower bounded of the calibration function in Theorem 3 seems to be non-trivial (I assume the results are correct). From the conclusion section, it also seems that these efforts do make sense in some cases. Negative comments: 1, According to the authors, the consistency problem of inconsistent surrogates is the main concern of this paper. However, it is not clear to me what motivates this study. That is, what motivates the study of consistency of inconsistent surrogates. I noticed some comments in the last section of the manuscript, which did not help me out. 2, The authors spent too much effort in stating the main results of the study of Osokin et al. [13]. Actually, 5 out of 8 papers were revisiting their results. After reading the manuscript, it makes me feel that the authors studied a mathematical problem that has not been solved in [13]. 3, This paper is extremely hard to follow. Many notions are not clearly defined. --- After reading the authors' feedback and the comments from the other two reviewers, I tend to vote for the acceptance of this paper.

Reviewer 3



The paper analyzes consistency properties of surrogate losses for structured prediction problems, generalizing the result of Osokin et al. to the inconsistent surrogates. The behavior of the proposed bound is illustrated in two concrete cases. Overall, the paper is technically solid and makes positive contribution. In the inconsistent case, a non-trivial bound on the calibration function is given, which is a reasonable contribution. However it is not discussed in the paper how better bounds would lead to algorithmic implications. In terms of presentation, the paper is also hard to follow, partly due to the fact that this work is heavily based on a previous paper (Osokin et al.). Additionally, the notation is overloaded and sometimes unclear. I believe the presentation can be largely improved. A few typos found: Line 104: minimize -> minimized Equation (6) the order of losses is not consistent with mentioned in text. Line 150 \eta -> \epsilon Line 156 than -> then